# A mathematical model of COVID-19 with multiple variants of the virus under optimal control in Ghana

**Young Rock Kim**[1]☯, **Youngho Min**[1]☯*, **Joy Nana Okogun-Odompley**[2]☯

**1** Major in Mathematics Education, Graduate School of Education, Hankuk University of Foreign Studies, Seoul, Republic of Korea, **2** Department of Mathematics, Hankuk University of Foreign Studies, Yongin, Republic of Korea

☯ These authors contributed equally to this work.
* mathmyh@khu.ac.kr

**Data Availability Statement:** All relevant files are available from the Zenodo repository (https://doi.org/10.5281/zenodo.11117064).

**Funding:** This work was supported by the Hankuk University of Foreign Studies Research Fund.

## Abstract

In this paper, we suggest a mathematical model of COVID-19 with multiple variants of the virus under optimal control. Mathematical modeling has been used to gain deeper insights into the transmission of COVID-19, and various prevention and control strategies have been implemented to mitigate its spread. Our model is a *SEIR*-based model for multi-strains of COVID-19 with 7 compartments. We also consider the circulatory structure to account for the termination of immunity for COVID-19. The model is established in terms of the positivity and boundedness of the solution and the existence of equilibrium points, and the local stability of the solution. As a result of fitting data of COVID-19 in Ghana to the model, the basic reproduction number of the original virus and Delta variant was estimated to be 1.9396, and the basic reproduction number of the Omicron variant was estimated to be 3.4905, which is 1.8 times larger than that. We observe that even small differences in the incubation and recovery periods of two strains with the same initial transmission rate resulted in large differences in the number of infected individuals. In the case of COVID-19, infections caused by the Omicron variant occur 1.5 to 10 times more than those caused by the original virus. In terms of the optimal control strategy, we formulate three control strategies focusing on social distancing, vaccination, and testing-treatment. We have developed an optimal control model for the three strategies outlined above for the multi-strain model using the Pontryagin's Maximum Principle. Through numerical simulations, we analyze three optimal control strategies for each strain and also consider combinations of the two control strategies. As a result of the simulation, all control strategies are effective in reducing disease spread, in particular, vaccination strategies are more effective than the other two control strategies. In addition the combination of the two strategies also reduces the number of infected individuals by 1/10 compared to implementing one strategy, even when mild levels are implemented. Finally, we show that if the testing-treatment strategy is not properly implemented, the number of asymptomatic and unidentified infections may surge. These results could help guide the level of government intervention and prevention strategy formulation.

Young Rock Kim and Youngho Min were supported by the National Research Foundation of Korea (NRF) grant funded by the Korea government (MSIT) (No. 2021 R1A2C1011467). The funders had no role in study design, data collection and analysis, decision to publish, or preparation of the manuscript.

**Competing interests:** The authors have declared that no competing interests exist.

## Introduction

The world came to a halt with the outbreak of severe acute respiratory syndrome (SARS) caused by SARS-CoV-2 that began in Wuhan, China [1] at the end of 2019. World Health Organization (WHO) declared the novel coronavirus (COVID-19) outbreak a global pandemic on March 11, 2020 [2]. Ghana recorded its first COVID-19 case on March 12, 2020. The two cases were revealed to be individuals who had returned to the country from Norway and Türkiye. These imported cases initiated the first contact tracing process in Ghana, helping detect several dozens of cases in a short period. A partial lockdown was issued by Ghana's President Nana Akufo-Addo in the Greater Accra region on March 28, 2020, and in the Kumasi Metropolitan area on March 30, 2020 [3]. As of December 4, 2022, Ghana was ranked 121st with the highest number of confirmed cases of 171,018 with 1,461 deaths, 5,279 casualties, and 169,541 recoveries [4].

Infectious diseases, such as COVID-19, that can lead to death often require mathematical modeling to understand the dynamics of transmission, and simulations can be used for predictive purposes, such as predicting trends in the number of infected individuals [5, 6]. The classical susceptible-infected-recovered (*SIR*) model was first proposed by William Ogilvy Kermack and Anderson Gray McKendrick and is known as the Kermack-McKendrick theory [7]. After that, a group of individuals who have not yet developed symptoms and are not infectious was introduced into a new compartment called Exposed (*E*), and a *SEIR* model with this compartment has been used for modeling various infectious diseases [8, 9]. Furthermore, models adjusted to specific disease scenarios have been developed, including models with vaccination [10, 11], models accounting for isolated individuals [12, 13], and even highly detailed *SIDARTHE* models [14, 15] (susceptible (*S*), infected (*I*), diagnosed (*D*), ailing (*A*), recognized (*R*), threatened (*T*), healed (*H*), and extinct (*E*)).

For COVID-19, Khajanchi et al. adapted the *SEIR* model to account for contact tracing and hospitalization strategies with regard to COVID-19 transmission [16]. They fitted this model using data from five states in India and demonstrated through model simulations that COVID-19 cases across India followed a power-law distribution by the end of September 2020. Samui et al. illustrated the effectiveness of disease transmission rate mitigation on the basic reproduction number and predicted the trend of infections using their COVID-19 model [17]. Previous studies on outbreaks [18–20] mainly focused on predicting the number of infected individuals tailored to the strategies and transmission situations of each country. Additionally, studies on optimal control have emerged to theoretically demonstrate the effectiveness of prevention measures, taking into account the introduction of various strategies and vaccines. Optimal control, previously integrated into various disease models such as leukaemia [21], Hepatitis B [22], Malaria [23, 24], HIV [25, 26], and COVID-19 [27, 28], has proven to be a successful tool in providing the right interventions and strategies to curb the spread of infectious diseases. Studies like [29–32] utilized optimal control theory in COVID-19 disease models to propose interventions and strategies.

However, with the emergence of mutations causing more harm and giving rise to new variants, the pandemic was facing a new situation. COVID-19 virus has mutated primarily into variants such as Alpha, Beta, Delta, and Omicron [33, 34]. As a result, multi-strain models of infectious disease COVID-19 were developed, which have proven to be the most useful for understanding the spread of infectious diseases [35, 36]. Researchers have investigated the multi-strain model, however, most of this research primarily focuses on the incidence rates [37, 38]. Khyar and Alali [33] investigated a two-strain epidemic model with incidence rates; they established the model in terms of the existence, positivity, and boundedness of solutions. They calculated four equilibrium points namely the disease-free equilibrium, the endemic

equilibrium with respect to strain 1, the endemic equilibrium with respect to strain 2, and the last endemic equilibrium with respect to both strains and it was observed that the model with generalized incidence functions contains many models with classical incidence functions and it gives a significant wide view of the equilibria stability.

In advanced studies on multiple strains, Kim, Choi, and Min. [39] investigated a multi-strain epidemic model which was based on *SEIR* model to simulate the COVID-19 pandemic. Their model, the *SVEIHRM* model, consisted of multiple vaccinations and mutant viruses of COVID-19. According to the antibody formation rate following multiple vaccinations, break-through infection was also considered in their model. As a result, they obtained the effective reproduction numbers of the original virus, the Delta, and the Omicron variants by fitting this model to data in Korea. Yaagoub and Allali [36] studied a three-strain *SEIR* epidemic model with a vaccination strategy with a system based on nine nonlinear ordinary differential equations. In their research, numerical simulations were used to confirm their theoretical results and it was shown that the basic reproduction number of all the strains must be less than the unity to eradicate the infection. However, these studies did not consider the optimal control strategies of their models.

Abioye et. al [27] proposed a mathematical model of COVID-19 in Nigeria with optimal control. The basic model was expanded into an optimal quarantine system, including three quarantine strategies: use of face masks and hand sanitizers along with social distancing, treatment of COVID-19 patients, and active screening of people who have recovered from COVID-19. The biological findings showed that COVID-19 could be effectively managed or eliminated in Nigeria if the control measures implemented are capable of taking or sustaining the basic reproduction number to a value below unity. Also, they opined that if the three control strategies are well managed by the NCDC, Presidential Task Force (PTF), and Federal Ministry of Health (FMOH) or policymakers COVID-19 in Nigeria will be eradicated. This study was limited in handling multiple strains because it focused on a single-strain model.

Most of the previous studies were conducted under the rapidly changing COVID-19 pandemic. Countries around the world faced an unprecedented emergency for the first time, and implemented various policies to cope with the crisis. However, with sufficient data on currently available diseases, our study motivation was to thoroughly investigate related data from the past in order to prevent the emergence of infectious diseases that threaten humanity, such as COVID-19 pandemic, and to minimize human damage through effective quarantine policies. We presented optimal control strategies for each strain using a multi-strain model and were able to analyze in detail the control strategies that were also implemented. This study proposes a multi-strain model considering the compartments model with quarantine (*SEIQR*) that has been prominently applied in existing research on COVID-19 models. Additionally, three control strategies—social distancing, vaccination, and testing-treatment—are applied to both the original and mutant viruses. We emphasize that our model is a multi-strain model, comprising two distinct strains. These properties allow us to apply optimal control measures comprehensively and analyze the impact of each strategy on the epidemiology of each strain separately. This will facilitate thorough investigation of the effectiveness of social distancing, vaccination, and testing-and-treating of COVID-19 patients for each specific variant. These details of our model enhance our ability to tailor and optimize control strategies in response to the dynamics of individual strains. Furthermore, we assess the effectiveness of controls by simulating the combination of two novel control strategies not previously considered. Based on these results, we suggest which strategies may be effective in dealing with newly emerging infectious diseases.

The paper is organized as follows. In Materials and methods section, we introduce a multi-strain model for COVID-19 constructed from a system of ordinary differential equations. We calculate the model's basic reproduction number and demonstrate the stability of the solution. For the optimal control problems, we show the uniqueness and existence of the solution. In result section, we estimate the transmission rates for each strain using actual data from Ghana using the least squares fitting. We employ these transmission rates to compute the basic reproduction number for each strain. Next, we compare the efficiency of the strategies using an optimal model. We discuss the efficiency of the strategies by measuring the number of infections occurring at the peak and the duration of reaching the peak in various situations. Throughout this process, we conducted a more nuanced data fitting to advance our understanding of COVID-19 dynamics. We look forward to using this study as a basis for considering more specific models for COVID-19.

## Materials and methods

In this section, we will introduce a multi-strain model for COVID-19 constructed from a system of ordinary differential equations. This model is based on the *SEIR* model, which includes compartments of exposed and infected individuals for each of the two strains and additionally includes a quarantine compartment.

### A multi-strain COVID-19 model

We denote the total population at time $t$ by $N(t)$ to construct a mathematical model using a system of ordinary differential equations. The total population, denoted as $N(t)$, is divided into 7 non-overlapping compartments. Therefore, individuals are allocated to 7 compartments representing susceptible individuals ($S(t)$), individuals exposed to strains 1 and 2 ($E_1(t)$, $E_2(t)$), individuals infected with strains 1 and 2 ($I_1(t)$, $I_2(t)$), quarantined individuals ($Q(t)$), and recovered individuals ($R(t)$). Therefore, $N(t) = S(t) + E_1(t) + E_2(t) + I_1(t) + I_2(t) + Q(t) + R(t)$. A detailed description of each compartment is provided below:

- $S$ (Susceptible): compartment for healthy individuals who can get exposed to diseases.

- $E_1$ (Exposed for strain 1): compartment of individuals from the susceptible group who get exposed to strain 1 ($I_1$). Individuals in this group can move to the infected compartment of strain 1.

- $E_2$ (Exposed for strain 2): compartment of individuals from the susceptible group who get exposed to strain 2 ($I_2$). Individuals in this group can move to the infected compartment of the strain2.

- $I_1$ (Infected of strain 1): Infected individuals of strain 1 come from those exposed to the original virus and Delta variant, they get infected with the virus and would further move into the quarantine compartment or recovered compartment from the disease without being quarantined.

- $A_1$ (Asymptomatic cases of strain 1 infection): compartment of individuals with asymptomatic or mild symptoms among individuals in $I_1$. Individuals in this compartment are not tested for COVID-19 because they do not show symptoms of infection. Therefore, it is converted to a recovered group ($R$) without quarantine measures.

- $I_2$ (Infected of strain 2): Infected individuals of strain 2 come from those exposed to the Omicron variant, they get infected with the virus and would further move into the quarantine compartment or recovered compartment from the disease without being quarantined.

- $A_2$ (Asymptomatic cases of strain 2 infection): compartment of individuals with asymptomatic or mild symptoms among individuals in $I_2$. Individuals in this compartment are not tested for COVID-19 because they do not show symptoms of infection. Therefore, it is converted to a recovered group ($R$) without quarantine measures.

- $Q$ (Quarantine): Individuals who get infected with the virus ($I_1$, $I_2$) get quarantined to reduce the spread of the disease and further moved to the recovered group.

- $R$ (Recovered): Individuals who get quarantined ($Q$) move to this group and get removed. Also, individuals from the infected group ($I_1$, $I_2$) could move to this group without being quarantined. The recovered individuals further move back into the susceptible group ($S$).

We can formulate the model by configuring the dynamics of the compartments defined above. Prior to this, we will present the assumptions underlying our model.

1. The SIR model typically assumes a closed population interval. However, as the COVID-19 we are addressing has persisted over an extended period, our model incorporates assumptions concerning births and natural deaths. Newborns are included in the susceptible group, and in each compartment, the population decreases at a rate corresponding to the mortality rate.

2. Our model excludes the simultaneous infection with two different strains, operating as a multi-strain model.

3. The exposed group ($E_i$) represents individuals who have been in contact with infected individuals of the $i$-th strain but have not exhibited symptoms, where $i$ = 1, 2. Individuals within this compartment transition to the infected groups ($I_i$) for each strain based on the transmission rate ($\beta_i$).

4. The infected groups ($I_i$) for each strain include both symptomatic and asymptomatic individuals, where $i$ = 1, 2. Symptomatic individuals identified in each strain are isolated and transition to the quarantine group at proportions determined by $\alpha_1$ and $\alpha_2$. Mildly symptomatic patients and asymptomatic individuals ($A_i$) within each strain do not undergo isolation and directly move to the recovered group ($R$) at a rate determined by $\gamma_i$.

5. In the infected groups for each strain ($I_1$ and $I_2$), we incorporate considerations for the reduction in population due to natural mortality, in addition to the mortality rate attributed to COVID-19.

6. Our model assumes the possibility of reinfection as the immunity period ends. Therefore, individuals who have recovered transition back to the susceptible group over time.

A diagram showing our model, incorporating the aforementioned compartments and assumptions, is presented in Fig 1. The arrows in the diagram represent the transitions of individuals between compartments. For example, individuals who are infected at time $t$ can transition to the compartment of quarantined individuals at time $t$ + 1. Using this property, we will establish a system of ordinary differential equations based on the rates of change for each compartment. Individuals belonging to the $S$ compartment are exposed to strain 1 (strain 2) of COVID-19 and move to $E_1$ ($E_2$), respectively. Individuals belonging to $E_1$ ($E_2$) develop disease over time and move to $I_1$ ($I_2$), respectively. Individuals in $I_1$ and $I_2$ are quarantined or die from COVID-19. However, some of them, either asymptomatic individuals or those with mild symptoms, go straight to compartment $R$ without being tested for COVID-19. Finally, individuals belonging to $R$ move back to the $S$ compartment when the antibody duration is over. The

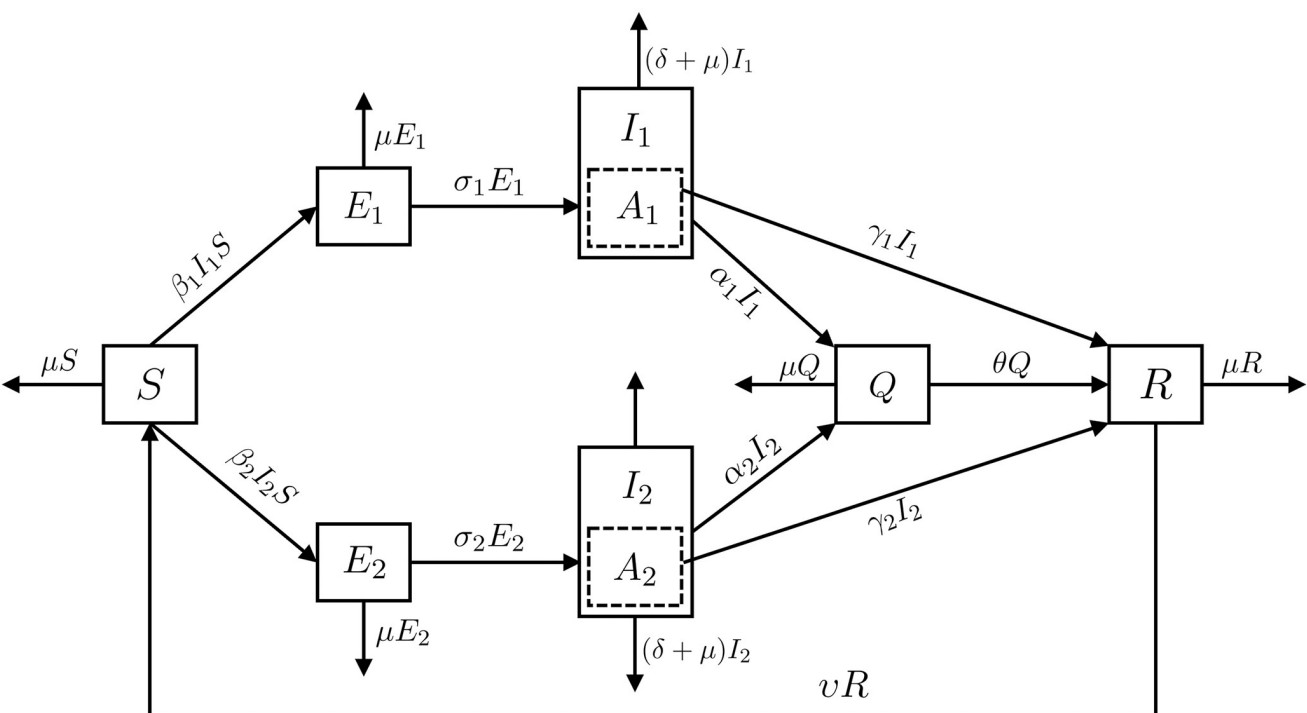

**Fig 1. Diagram of the 2-strain COVID-19 model.**

transfer rate parameters used in the model are listed in Table 1. In this dynamic system, all parameters in Table 1 are nonnegative.

Also, taking into account the birth rate and natural death rate additionally, the rate of

change in $S$ over time is represented by $\Lambda - \dfrac{\beta_1 I_1 S}{N} - \dfrac{\beta_2 I_2 S}{N} - \mu S + \upsilon R$. Similarly, if we

**Table 1. Parameters of the model.**

| Parameter | Description |
|---|---|
| $\Lambda$ | Birth rate |
| $\beta_1, \beta_2$ | Transmission rate for strain 1, 2 |
| $\sigma_1, \sigma_2$ | Reciprocal of incubation period for strain 1, 2 |
| $\alpha_1, \alpha_2$ | Detection rate of infected individuals for strain 1, strain 2 |
| $\gamma_1, \gamma_2$ | Reciprocal of the recovery period of asymptomatic or mildly symptomatic individuals in $A_1$ and $A_2$ |
| $\theta$ | Recovery rate for individuals in $Q$ |
| $\mu$ | Natural death rate |
| $\delta$ | COVID-19 induced death rate |
| $\upsilon$ | Rate of recovered individuals returning to a susceptible (Reciprocal of antibody formation period) |

Table notes the parameters used in the model and their descriptions.

combine the rates of change over time for the other compartments, we get the following model.

$$\frac{dS}{dt} = \Lambda - \frac{\beta_1 I_1 S}{N} - \frac{\beta_2 I_2 S}{N} - \mu S + \upsilon R,$$

$$\frac{dE_1}{dt} = \frac{\beta_1 I_1 S}{N} - (\mu + \sigma_1)E_1,$$

$$\frac{dE_2}{dt} = \frac{\beta_2 I_2 S}{N} - (\mu + \sigma_2)E_2,$$

$$\frac{dI_1}{dt} = \sigma_1 E_1 - (\delta + \mu + \alpha_1 + \gamma_1)I_1, \tag{1}$$

$$\frac{dI_2}{dt} = \sigma_2 E_2 - (\delta + \mu + \alpha_2 + \gamma_2)I_2,$$

$$\frac{dQ}{dt} = \alpha_1 I_1 + \alpha_2 I_2 - (\mu + \theta)Q,$$

$$\frac{dR}{dt} = \theta Q + \gamma_1 I_1 + \gamma_2 I_2 - (\upsilon + \mu)R.$$

Expressing each of the 7 compartments as the fraction of the total population $N$ (in other words, $S = \frac{S}{N}$, $E_i = \frac{E_i}{N}$, $I_i = \frac{I_i}{N}$, $Q = \frac{Q}{N}$, and $R = \frac{R}{N}$ for $i = 1, 2$), then we obtain

$$\frac{dS}{dt} = \Lambda - \beta_1 I_1 S - \beta_2 I_2 S - \mu S + \upsilon R,$$

$$\frac{dE_1}{dt} = \beta_1 I_1 S - aE_1,$$

$$\frac{dE_2}{dt} = \beta_2 I_2 S - bE_2,$$

$$\frac{dI_1}{dt} = \sigma_1 E_1 - cI_1, \tag{2}$$

$$\frac{dI_2}{dt} = \sigma_2 E_2 - dI_2,$$

$$\frac{dQ}{dt} = \alpha_1 I_1 + \alpha_2 I_2 - (\mu + \theta)Q,$$

$$\frac{dR}{dt} = \theta Q + \gamma_1 I_1 + \gamma_2 I_2 - (\upsilon + \mu)R.$$

where $a = (\mu + \sigma_1)$, $b = (\mu + \sigma_2)$, $c = (\delta + \mu + \alpha_1 + \gamma_1)$, and $d = (\delta + \mu + \alpha_2 + \gamma_2)$, We denote the state vector $(S(t), E_1(t), E_2(t), I_1(t), I_2(t), Q(t), R(t))$ as $X(t)$. Then equations in (2) describe the state evolution of the model with state vector $X(t)$ as the following nonlinear dynamic system which involves the nonlinear terms $I_1 S$ and $I_2 S$:

$$\frac{dX(t)}{dt} = A(X, t)X(t) + \Lambda e_1, \tag{3}$$

where

$$
A(X, t) = \begin{pmatrix}
-\beta_1 I_1 - \beta_2 I_2 - \mu & 0 & 0 & 0 & 0 & 0 & \upsilon \\
0 & -a & 0 & \beta_1 S & 0 & 0 & 0 \\
0 & 0 & -b & 0 & \beta_2 S & 0 & 0 \\
0 & \sigma_1 & 0 & -c & 0 & 0 & 0 \\
0 & 0 & \sigma_2 & 0 & -d & 0 & 0 \\
0 & 0 & 0 & \alpha_1 & \alpha_2 & -(\mu + \theta) & 0 \\
0 & 0 & 0 & \gamma_1 & \gamma_2 & \theta & -(\upsilon + \mu)
\end{pmatrix},
$$

$e_1$ is the unit 7th dimensional Euclidean vector with first component being one.

## Model analysis

Starting with the proof of the positivity and boundedness of the solutions, we will conduct a quantitative study including invariant region and equilibrium points in our model. In addition, we will show the local stability of the model and find the basic reproduction number.

**Positivity and boundedness of the solution.** To show the non-negativeness of the solution of Eq (3), we use the theory (Lemma 1) from reference [40].

**Lemma 1**. *If $A(X, t)$ is a Metzler matrix (i.e. all of its off-diagonal entries are non-negative) and $\Lambda$ is a non-negative constant, then the state trajectory solution of* Eq (3) *is non-negative for all t.*

We can easily prove the following Theorem 1 using this Lemma 1.

**Theorem 1**. *For all non-negative initial conditions, the solutions of the model* (2) *are non-negative.*

*Proof.* Because of Lemma 1, we just need to show that matrix $A(X, t)$ is a Metzler matrix for all $t$. Since all parameters defined in Table 1 are non-negative, then $A(X, t)$ is a Metzler.

**Theorem 2**. *For all non-negative initial conditions, the sum of all elements of $X(t)$ is bounded.*

*Proof.* Note that total human population $N(t)$ is the sum of all elements of $X(t)$. From model Eq (2), we obtain

$$
\frac{dN(t)}{dt} = \Lambda - \mu N(t) - \delta(I_1 + I_2). \tag{4}
$$

By Theorem 1, we have the following inequality,

$$
\Lambda - (\mu + \delta)N(t) \leq \frac{dN(t)}{dt} \leq \Lambda - \mu N(t). \tag{5}
$$

This can be rewritten as

$$
\frac{\Lambda}{\mu + \delta} \leq \liminf_{t \to \infty} N(t) \leq \limsup_{t \to \infty} N(t) \leq \frac{\Lambda}{\delta}. \tag{6}
$$

Therefore $N(t)$ is bounded.

**Invariant region of COVID-19 model.** According to Peter et. al. [13], the invariant region can be defined as the domain in which solutions to the model are of both biological and mathematical importance.

**Theorem 3**. *The region $\Omega \subset \mathbb{R}_+^7$ is a non-negative invariant for the model* (2) *with seven non-negative initial conditions.*

*Proof.* Let $\Omega$ be the feasible region of the COVID-19 model (2). To say that the domain $\Omega$ is invariant, all solutions $X(t)$ in Eq (2) again fall into the domain $\Omega$ for all $t$. By Theorem 1 and 2,

we can easily know that all elements of $X(t)$ are less than or equal to $N(t)$. Therefore, $X(t)$ belongs to the domain $\Omega$.

The feasible region of the model (2) can be expressed as $\Omega \subset \mathbb{R}^7_+$ with

$$\Omega = \{X(t) \in \mathbb{R}^7_+ \mid S + E_1 + E_2 + I_1 + I_2 + Q + R \le \frac{\Lambda}{\mu}\}.$$

**Basic reproduction numbers.** The basic reproduction number ($R_0$) is an epidemiological notion to measure the spread of an infectious disease [41]. The next-generation matrix $FV^{-1}$ would be used to calculate $R_0$ [42, 43]. A detailed formula derivation process is attached in S1 Appendix.

The reproduction number $R_0 = \rho(FV^{-1})$, where $\rho$ is the spectral radius, $F$ is the non-negative matrix of new infection cases and $V$ is the matrix of the transition of infections associated with the model. In our model, the $F$ and $V$ matrices are:

$$F = \begin{pmatrix} 0 & 0 & \beta_1 & 0 \\ 0 & 0 & 0 & \beta_2 \\ 0 & 0 & 0 & 0 \\ 0 & 0 & 0 & 0 \end{pmatrix}, \quad V = \begin{pmatrix} a & 0 & 0 & 0 \\ 0 & b & 0 & 0 \\ -\sigma_1 & 0 & c & 0 \\ 0 & -\sigma_2 & 0 & d \end{pmatrix} \tag{7}$$

where $a = \mu + \sigma_1$, $b = \mu + \sigma_2$, $c = \delta + \mu + \alpha_1 + \gamma_1$, $d = \delta + \mu + \alpha_2 + \gamma_2$. Hence we get the next-generation matrix

$$FV^{-1} = \begin{pmatrix} \frac{\beta_1 \sigma_1}{ac} & 0 & \frac{\beta_1}{c} & 0 \\ 0 & \frac{\beta_2 \sigma_2}{bd} & 0 & \frac{\beta_2}{d} \\ 0 & 0 & 0 & 0 \\ 0 & 0 & 0 & 0 \end{pmatrix}. \tag{8}$$

Therefore, the basic reproduction number is given by

$$R_0 = \max(R_0^1, R_0^2) \tag{9}$$

with

$$R_0^1 = \frac{\beta_1 \sigma_1}{ac} \quad \text{and} \quad R_0^2 = \frac{\beta_2 \sigma_2}{bd}. \tag{10}$$

Here, we can interpret $R_0^1$ as the reproduction number for strain 1 and $R_0^2$ as the reproduction number for strain 2.

**Existence of the equilibrium points.** The COVID-19 model with two strains has three equilibria: in other words, the disease-free equilibrium ($\varepsilon_0$), the endemic for strain 1 ($\varepsilon_1$), and the endemic for strain 2 ($\varepsilon_2$). Since the first six equations are independent of $R$, we can omit

the seventh equation of (2) and the problem would be reduced to:

$$\frac{dS}{dt} = \Lambda - \beta_1 I_1 S - \beta_2 I_2 S - \mu S + \upsilon R,$$

$$\frac{dE_1}{dt} = \beta_1 I_1 S - (\mu + \sigma_1)E_1,$$

$$\frac{dE_2}{dt} = \beta_2 I_2 S - (\mu + \sigma_2)E_2,$$

$$\frac{dI_1}{dt} = \sigma_1 E_1 - (\delta + \mu + \alpha_1 + \gamma_1)I_1,$$

$$\frac{dI_2}{dt} = \sigma_2 E_2 - (\delta + \mu + \alpha_2 + \gamma_2)I_2,$$

$$\frac{dQ}{dt} = \alpha_1 I_1 + \alpha_2 I_2 - (\mu + \theta)Q$$

(11)

with $R = N - S - E_1 - E_2 - I_1 - I_2 - Q$.

**Theorem 4**. *The model* Eq (11) *have the disease-free equilibrium ($\varepsilon_0$) and two endemic equilibria ($\varepsilon_1$) and ($\varepsilon_2$). Hence,*

- *The strain 1 endemic equilibrium ($\varepsilon_1$) exists when $R_0^1 > 1$*

- *The strain 2 endemic equilibrium ($\varepsilon_2$) exists when $R_0^2 > 1$*

*Proof*. In order to get the equilibria, we solve Eq (11) to obtain $\varepsilon_0, \varepsilon_1$ and $\varepsilon_2$ A detailed proof of theorem 4 is given in S1 Appendix.

- When $I_1 = 0$ and $I_2 = 0$, the disease-free equilibrium (DFE) of the model
$$\varepsilon_0 = \left(\frac{\Lambda}{\mu}, 0, 0, 0, 0, 0\right)$$

- When $I_1 \neq 0$ and $I_2 = 0$, the strain 1 endemic equilibrium
$$\varepsilon_1 = \left(S_1^*, \frac{1}{a}(\Lambda - \mu S_1^*), 0, \frac{\sigma_1}{ac}(\Lambda - \mu S_1^*), 0, \frac{\alpha_1 \sigma_1}{ac(\mu + \theta)}(\Lambda - \mu S_1^*)\right)$$

- When $I_1 = 0$ and $I_2 \neq 0$, the strain 2 endemic equilibrium
$$\varepsilon_2 = \left(S_2^*, 0, \frac{1}{b}(\Lambda - \mu S_2^*), 0, \frac{\sigma_2}{bd}(\Lambda - \mu S_2^*), \frac{\alpha_2 \sigma_2}{bd(\mu + \theta)}(\Lambda - \mu S_2^*)\right)$$

where $a = \mu + \sigma_1, b = \mu + \sigma_2, c = \delta + \mu + \alpha_1 + \gamma_1, d = \delta + \mu + \alpha_2 + \gamma_2, S_1^* = \dfrac{\Lambda}{\beta_1 I_1 + \mu}$, and

$S_2^* = \dfrac{\Lambda}{\beta_2 I_2 + \mu}$.

**Local stability.**   To obtain the local stability, the system of Eq (2) is linearized to estimate the Jacobian matrix of $f_1, f_2 \cdots, f_7$ where

$$
\begin{aligned}
f_1 &= \frac{dS}{dt} = \Lambda - \beta_1 I_1 S - \beta_2 I_2 S - \mu S + \upsilon R,\\[4pt]
f_2 &= \frac{dE_1}{dt} = \beta_1 I_1 S - (\mu + \sigma_1)E_1,\\[4pt]
f_3 &= \frac{dE_2}{dt} = \beta_2 I_2 S - (\mu + \sigma_2)E_2,\\[4pt]
f_4 &= \frac{dI_1}{dt} = \sigma_1 E_1 - (\delta + \mu + \alpha_1 + \gamma_1)I_1,\\[4pt]
f_5 &= \frac{dI_2}{dt} = \sigma_2 E_2 - (\delta + \mu + \alpha_2 + \gamma_2)I_2,\\[4pt]
f_6 &= \frac{dQ}{dt} = \alpha_1 I_1 + \alpha_2 I_2 - (\mu + \theta)Q,\\[4pt]
f_7 &= \frac{dR}{dt} = \theta Q + \gamma_1 I_1 + \gamma_2 I_2 - (\upsilon + \mu)R.
\end{aligned}
\tag{12}
$$

Then we obtain the next Jacobian matrix.

$$
J = \begin{pmatrix}
-\beta_1 I_1 - \beta_2 I_2 - \mu & 0 & 0 & -\beta_1 S & -\beta_2 S & 0 & \upsilon \\
\beta_1 I_1 & -a & 0 & \beta_1 S & 0 & 0 & 0 \\
\beta_2 I_2 & 0 & -b & 0 & \beta_2 S & 0 & 0 \\
0 & \sigma_1 & 0 & -c & 0 & 0 & 0 \\
0 & 0 & \sigma_2 & 0 & -d & 0 & 0 \\
0 & 0 & 0 & \alpha_1 & \alpha_2 & -(\mu + \theta) & 0 \\
0 & 0 & 0 & \gamma_1 & \gamma_2 & \theta & -(\upsilon + \mu)
\end{pmatrix}.
\tag{13}
$$

To determine the local stability of the disease-free equilibrium, we evaluate the Jacobian at $\varepsilon_0$. At the DFE, $S = 1, E_1 = 0, E_2 = 0, I_1 = 0, I_2 = 0, Q = 0, R = 0$, then

$$
J(\varepsilon_0) = \begin{pmatrix}
-\mu & 0 & 0 & -\beta_1 & -\beta_2 & 0 & \upsilon \\
0 & -a & 0 & \beta_1 & 0 & 0 & 0 \\
0 & 0 & -b & 0 & \beta_2 & 0 & 0 \\
0 & \sigma_1 & 0 & -c & 0 & 0 & 0 \\
0 & 0 & \sigma_2 & 0 & -d & 0 & 0 \\
0 & 0 & 0 & \alpha_1 & \alpha_2 & -(\mu + \theta) & 0 \\
0 & 0 & 0 & \gamma_1 & \gamma_2 & \theta & -(\upsilon + \mu)
\end{pmatrix}.
\tag{14}
$$

Recall that $a = \mu + \sigma_1, b = \mu + \sigma_2, c = \delta + \mu + \alpha_1 + \gamma_1, d = \delta + \mu + \alpha_2 + \gamma_2$. Using MATLAB, the

eigenvalues evaluated are as follows:

$$\Lambda_1 = -(\mu + \theta), \ \Lambda_2 = -(\mu + \upsilon),$$
$$\Lambda_3 = -\mu, \ \ \Lambda_4 = -(c + \sqrt{\beta_1 \sigma_1}),$$
$$\Lambda_5 = (-a + \sqrt{\beta_1 \sigma_1}), \ \ \Lambda_6 = -(d + \sqrt{\beta_2 \sigma_2}),$$
$$\Lambda_7 = (-b + \sqrt{\beta_2 \sigma_2}).$$

Since $S = 1$ for the disease-free equilibrium, the eigenvalues are negative if and only if $R_0 < 1$. Therefore, the local stability of the disease-free equilibrium is locally stable.

**Theorem 5**. *The endemic equilibrium $\varepsilon_1$ of strain 1 is locally asymptotically stable if $R_0^1 \geq 1$, otherwise unstable.*

*Proof*. At the endemic state of strain 1, $I_1 \neq 0$ and $I_2 = 0$. The Jacobian matrix $J(\varepsilon_1)$ is evaluated from Eq (13) as

$$J(\varepsilon_1) = \begin{pmatrix} -\beta_1 I_1 - \mu & 0 & 0 & -\beta_1 S & -\beta_2 S & 0 & \upsilon \\ \beta_1 I_1 & -a & 0 & \beta_1 S & 0 & 0 & 0 \\ 0 & 0 & -b & 0 & \beta_2 S & 0 & 0 \\ 0 & \sigma_1 & 0 & -c & 0 & 0 & 0 \\ 0 & 0 & \sigma_2 & 0 & -d & 0 & 0 \\ 0 & 0 & 0 & \alpha_1 & \alpha_2 & -(\mu + \theta) & 0 \\ 0 & 0 & 0 & \gamma_1 & \gamma_2 & \theta & -(\upsilon + \mu) \end{pmatrix}. \tag{15}$$

Using MATLAB, the eigenvalues evaluated are as follows:

$$\Lambda_1 = -(\mu + \theta), \ \ \Lambda_2 = -(d + \sqrt{S\beta_2 \sigma_2}), \ \Lambda_3 = (\sqrt{S\beta_2 \sigma_2} - b).$$

$\Lambda_4, \Lambda_5, \Lambda_6, \Lambda_7$ can be calculated using the characteristic equation of $J(\varepsilon_1)$,

$$\{(b + x)(d + x) - \sigma_2 \beta_2 S)(-\mu - \theta - x)\} \times$$
$$\{(-\upsilon - \mu - x)(\beta_1 I_1 - \mu - x)(-a - x)(-c - x) + \sigma_1 \beta_2 S(-\upsilon - \mu - x)(\beta_1 I_1 - \mu - x)$$
$$+ (-\upsilon - \mu - x)(\beta_1 I_1 \sigma_1)(-\beta_1 S) + \beta_1 I_1 \sigma_1 (-\upsilon \gamma_1)\} = 0.$$

Since the eigenvalues are negative, the endemic equilibrium of strain 1 is locally asymptotically stable.

**Theorem 6**. *The endemic equilibrium of strain 2 $\varepsilon_2$ is locally asymptotically stable if $R_0^2 \geq 1$, otherwise unstable.*

*Proof.* At the endemic state of strain 2, $I_1 = 0$ and $I_2 \neq 0$. The Jacobian matrix $J(\varepsilon_2)$ is evaluated from Eq (13) as

$$
J(\varepsilon_2) =
\begin{pmatrix}
-\beta_2 I_2 - \mu & 0 & 0 & -\beta_1 S & -\beta_2 S & 0 & \upsilon \\
0 & -a & 0 & \beta_1 S & 0 & 0 & 0 \\
\beta_2 I_2 & 0 & -b & 0 & \beta_2 S & 0 & 0 \\
0 & \sigma_1 & 0 & -c & 0 & 0 & 0 \\
0 & 0 & \sigma_2 & 0 & -d & 0 & 0 \\
0 & 0 & 0 & \alpha_1 & \alpha_2 & -(\mu + \theta) & 0 \\
0 & 0 & 0 & \gamma_1 & \gamma_2 & \theta & -(\upsilon + \mu)
\end{pmatrix}.
\tag{16}
$$

Using MATLAB, the eigenvalues evaluated were as follows:

$$
\Lambda_1 = -(\mu + \theta), \ \Lambda_2 = -(c + \sqrt{S\beta_1\sigma_1}), \ \Lambda_3 = (\sqrt{S\beta_1\sigma_1} - a).
$$

$\Lambda_4, \Lambda_5, \Lambda_6, \Lambda_7$ can be calculated using the characteristic equation of $J(\varepsilon_2)$,

$$
\begin{aligned}
& \sigma_1\alpha_2\beta_2 S(\beta_2 I_2 - \mu - x)(-a - x)(-\upsilon - \mu - x)(-c - x) \\
& -\sigma_1\beta_1 S(\beta_2 I_2 - \mu - x)(-a - x)(-\upsilon - \mu - x)(-\mu - \theta - x)\{(-b - x)(-d - x) - \beta_2 S\sigma_2\} \\
& -\beta_2 I_2\sigma_2\upsilon\gamma_2(-\mu - \theta - x)(-a - x)(-c - x) + \sigma_2\upsilon\gamma_2\beta_2 I_2\beta_1 S\sigma_1(-\mu - \theta - x) \\
& -\beta_2 S\sigma_2\beta_2 I_2(-\mu - \theta - x)(-\upsilon - \mu - x)(-a - x)(-c - x) \\
& -\beta_2 I_2\beta_2 S\beta_1 S\sigma_1(-\mu - \theta - x)(-\upsilon - \mu - x) = 0.
\end{aligned}
$$

Since the eigenvalues are negative, the endemic equilibrium of strain 2 is locally asymptotically stable.

## Analysis for optimal control of the model

Three control measures are incorporated into the multi-strain model. These measures include the social distancing ($u_1$), vaccination ($u_2$), and testing-treatment of COVID-19 patients ($u_3$). Therefore, COVID-19 reduction rate is $(1 - u_1)$ and the new infection rates are $(1 - u_1)\beta_1 I_1$ and $(1 - u_1)\beta_2 I_2$. The proposed optimal control of the multi-strain COVID-19 model is as

follows:

$$\frac{dS}{dt} = \Lambda - (1 - u_1)\beta_1 I_1 S - (1 - u_1)\beta_2 I_2 S - \mu S + \upsilon R - u_2 S,$$

$$\frac{dE_1}{dt} = (1 - u_1)\beta_1 I_1 S - (\mu + \sigma_1)E_1,$$

$$\frac{dE_2}{dt} = (1 - u_1)\beta_2 I_2 S - (\mu + \sigma_2)E_2,$$

$$\frac{dI_1}{dt} = \sigma_1 E_1 - (\delta + \mu + \alpha_1 u_3 + \gamma_1)I_1, \qquad (17)$$

$$\frac{dI_2}{dt} = \sigma_2 E_2 - (\delta + \mu + \alpha_2 u_3 + \gamma_2)I_2,$$

$$\frac{dQ}{dt} = \alpha_1 u_3 I_1 + \alpha_2 u_3 I_2 - (\mu + \theta)Q,$$

$$\frac{dR}{dt} = \theta Q + \gamma_1 I_1 + \gamma_2 I_2 - (\upsilon + \mu)R + u_2 S.$$

**Optimal control description.** The objective function of the system (17) is used to minimize the total number of exposed humans and infected humans using the control variables $u_1$, $u_2$, $u_3$. Also, all the control variables are non-negative. The objective function is defined as:

$$J = \int_0^{t_f} \left( A_1 E_1(t) + A_2 E_2(t) + A_3 I_1(t) + A_4 I_2(t) + \frac{1}{2}(P_1 u_1^2(t) + P_2 u_2^2(t) + P_3 u_3^2(t) \right) dt. \quad (18)$$

Subject to the system (17), where $A_1$ and $A_2$ are positive constant weights of exposed and $A_3$ and $A_4$ are the positive constant weights of infected humans. $P_1$, $P_2$ and $P_3$ are the positive constant weights of control variables $u_1$, $u_2$, and $u_3$ respectively while $P_1 u_1^2$, $P_2 u_2^2$ and $P_3 u_3^2$ are the quadratic costs associated with the social distancing ($u_1$), vaccination ($u_2$), and testing-treatment of COVID-19 patients ($u_3$) respectively. This quadratic cost and objective function were chosen in reference to the literature on epidemic controls by [24, 27]. We intend to find an optimal control of $u_1^*$, $u_2^*$ and $u_3^*$ such that

$$J(u_1^*, u_2^*, u_3^*) = \min\{(u_1, u_2, u_3) : u_1, u_2, u_3 \in \Omega\}, \qquad (19)$$

where $\Omega = \{u_i : 0 \leq u_i(t) \leq 1,$ Lebesgue measurable, $t \in [0, t_f]$ for $i = 1, 2, 3\}$ is the control set subject to optimal control model (17) with initial conditions.

**Existence of the optimal control.** With initial conditions at $t = 0$, the existence of the optimal control was proven and the properties of the model (17) were analyzed with all non-negative initial conditions for all positive $t > 0$. Using the optimal control in the system (17) to see the existence of optimal control with the necessary conditions satisfying the Pontryagin's Maximum Principle [44]. Pontryagin's Maximum Principle was applied to convert Eqs (17)–(19) into a problem of minimizing pointwise Lagrangian of the control problem, $L$, with respect to $u_1$, $u_2$, and $u_3$. The Lagrangian is given by:

$$L = A_1 E_1(t) + A_2 E_2(t) + A_3 I_1(t) + A_4 I_2(t) + \frac{1}{2}\left(P_1 u_1^2(t) + P_2 u_2^2(t) + P_3 u_3^2(t)\right). \qquad (20)$$

This would be used to find the minimal value of the Lagrangian. It could be achieved by considering Hamiltonian $H$ for the control problem as

$$H = L + \lambda_1 \frac{dS(t)}{dt} + \lambda_2 \frac{dE_1(t)}{dt} + \lambda_3 \frac{dE_2(t)}{dt} + \lambda_4 \frac{dI_1(t)}{dt} + \lambda_5 \frac{dI_2(t)}{dt} + \lambda_6 \frac{dQ(t)}{dt} + \lambda_7 \frac{dR(t)}{dt}. \qquad (21)$$

By substituting Eq (20) and model (17) into Eq (21), we obtain

$$
\begin{aligned}
H ={}& A_1 E_1(t) + A_2 E_2(t) + A_3 I_1(t) + A_4 I_2(t) + \frac{1}{2}\left[P_1 u_1^2(t) + P_2 u_2^2(t) + P_3 u_3^2(t)\right] \\
&+ \lambda_1[\Lambda - (1 - u_1(t))\beta_1 I_1 S - (1 - u_1(t))\beta_2 I_2 S - \mu S + \upsilon R + u_2(t)S] \\
&+ \lambda_2[(1 - u_1(t))\beta_1 I_1 S - (\mu + \sigma_1)E_1] + \lambda_3[(1 - u_1(t))\beta_2 I_2 S - (\mu + \sigma_2)E_2] \\
&+ \lambda_4[\sigma_1 E_1 - (\delta + \mu + \alpha_1 u_3(t) + \gamma_1)I_1] + \lambda_5[\sigma_2 E_2 - (\delta + \mu + \alpha_2 u_3(t) + \gamma_2)I_2] \\
&+ \lambda_6[\alpha_1 u_3(t)I_1 + \alpha_2 u_3(t)I_2 - (\mu + \theta)Q] + \lambda_7[\theta Q + \gamma_1 I_1 + \gamma_2 I_2 - (\upsilon + \mu)R]
\end{aligned}
\tag{22}
$$

where $\lambda_1$, $\lambda_2$, $\lambda_3$, $\lambda_4$, $\lambda_5$, $\lambda_6$ and $\lambda_7$ are the adjoint variables.

**Uniqueness of the optimal control.** Pontryagin's Maximum Principle would be utilized to determine the conditions for optimal control. This is done by minimizing the cost function in Eq (18) and subjecting it to the model (17).

$$
\begin{aligned}
\frac{dx}{dt} &= \frac{\partial H(t, x, u, \lambda)}{\partial \lambda}, \\
0 &= \frac{\partial H(t, x, u, \lambda)}{\partial u}, \\
\lambda' &= \frac{\partial H(t, x, u, \lambda)}{\partial x}.
\end{aligned}
\tag{23}
$$

where $u$ is for the control and $x$ is the associated state variable to minimize the given objective.

**Theorem 7.** *Let* $S^*$, $E_1^*$, $E_2^*$, $I_1^*$, $I_2^*$, $Q^*$, $R^*$ *be optimal state solutions associated with optimal control* $(u_1^*, u_2^*, u_3^*)$ *for the optimal control problem in model* (17) *and* Eq (18). *There exist the co-states* $\lambda_i$ *which verify* Eq (24) *with the transversality conditions* $\lambda_i(t_f) = 0$ *in* Eq (25) *for* $i = 1, 2, \cdots, 7$ *and in* Eq (27) *the control variables* $(u_1^*, u_2^*, u_3^*)$.

*Proof.* Let us differentiate Hamiltonian ($H$) (22) with respect to $S$, $E_1$, $E_2$, $I_1$, $I_2$, $Q$ and $R$. Furthermore, considering the state variables by applying the first and third equations in Eq (23) into Eq (22), we have the following:

$$
\begin{aligned}
\frac{d\lambda_1}{dt} &= -\frac{\partial H}{\partial S} = (1 - u_1)\beta_1 I_1(d\lambda_1 - \lambda_2) + (1 - u_1)\beta_2 I_2(\lambda_1 - \lambda_3) + \mu\lambda_1 + u_2(\lambda_1 - \lambda_7), \\
\frac{d\lambda_2}{dt} &= -\frac{\partial H}{\partial E_1} = \sigma_1(\lambda_2 - \lambda_4) + \mu\lambda_2 - A_1, \\
\frac{d\lambda_3}{dt} &= -\frac{\partial H}{\partial E_2} = \sigma_2(\lambda_3 - \lambda_5) + \mu\lambda_3 - A_2, \\
\frac{d\lambda_4}{dt} &= -\frac{\partial H}{\partial I_1} = (1 - u_1)\beta_1 S(\lambda_1 - \lambda_2) + \alpha_1 u_3(\lambda_4 - \lambda_6) + (\delta + \mu)\lambda_4 + \gamma_1(\lambda_4 - \lambda_7) - A_3, \quad (24) \\
\frac{d\lambda_5}{dt} &= -\frac{\partial H}{\partial I_2} = (1 - u_1)\beta_2 S(\lambda_1 - \lambda_3) + \alpha_2 u_3(\lambda_5 - \lambda_6) + (\delta + \mu)\lambda_5 + \gamma_2(\lambda_5 - \lambda_7) - A_4, \\
\frac{d\lambda_6}{dt} &= -\frac{\partial H}{\partial Q} = \theta(\lambda_6 - \lambda_7) + \mu\lambda_6, \\
\frac{d\lambda_7}{dt} &= -\frac{\partial H}{\partial R} = \mu\lambda_7 - \upsilon(\lambda_1 - \lambda_7).
\end{aligned}
$$

with the transversality conditions

$$
\lambda_1(t_f) = \lambda_2(t_f) = \lambda_3(t_f) = \lambda_4(t_f) = \lambda_5(t_f) = \lambda_6(t_f) = \lambda_7(t_f) = 0.
\tag{25}
$$

Evaluating the optimal control of the control variable set where $u_i = (0, 1)$, let $S = S^*$, $E_1 = E_1^*$,

$E_2 = E_2^*$, $I_1 = I_1^*$, $I_2 = I_2^*$, $Q = Q^*$ and $R = R^*$ and applying the second equation in Eq (23) by differentiating Hamiltonian, $H$ (22) with respect to the control variables $u_1$, $u_2$ and $u_3$ we obtain

$$\frac{\partial H}{\partial u_1} = P_1 u_1^* - \beta_1 I_1^* S^* (\lambda_2 - \lambda_1) - \beta_2 I_2^* S^* (\lambda_3 - \lambda_1) = 0,$$

$$\frac{\partial H}{\partial u_2} = P_2 u_2^* - S^* (\lambda_7 - \lambda_1) = 0, \tag{26}$$

$$\frac{\partial H}{\partial u_3} = P_3 u_3^* - \alpha_1 I_1^* (\lambda_6 - \lambda_4) - \alpha_2 I_2^* (\lambda_6 - \lambda_5) = 0.$$

If we solve (26), then we have

$$u_1^* = \frac{\beta_1 I_1^* S^* (\lambda_2 - \lambda_1) + \beta_2 I_2^* S^* (\lambda_3 - \lambda_1)}{P_1},$$

$$u_2^* = \frac{S^* (\lambda_7 - \lambda_1)}{P_2},$$

$$u_3^* = \frac{\alpha_1 I_1^* (\lambda_6 - \lambda_4) + \alpha_2 I_2^* (\lambda_6 - \lambda_5)}{P_3}.$$

Therefore, we have

$$u_1^* = \max \left\{ 0, \min \left( \frac{\beta_1 I_1^* S^* (\lambda_2 - \lambda_1) + \beta_2 I_2^* S^* (\lambda_3 - \lambda_1)}{P_1} \right) \right\},$$

$$u_2^* = \max \left\{ 0, \min \left( \frac{S^* (\lambda_7 - \lambda_1)}{P_2} \right) \right\}, \tag{27}$$

$$u_3^* = \max \left\{ 0, \min \left( \frac{\alpha_1 I_1^* (\lambda_6 - \lambda_4) + \alpha_2 I_2^* (\lambda_6 - \lambda_5)}{P_3} \right) \right\}.$$

## Results

This section is divided into two subsections. First, using the least squares fitting, we estimate the transmission rates for each strain using actual data from Ghana. With the obtained transmission rate, we calculate the basic reproduction number for each strain according to Eq (9). Next, we compare the efficiency of three strategies using an optimal control model (Eq (17)). The three strategies include preventive measures such as social distancing, vaccination, and testing-treatment of COVID-19 patients. Given that our model consists of two strains, we can analyze the efficiency of each strategy for each strain separately. We conducted the simulation using MATLAB, and the data and code can be found in the attached S1 File.

### Model fitting

To analyze Ghana's COVID-19 historical data, we retrieve the data from the World Health Organization dashboard [45]. The data from January 2020 to January 2023 is used for the analysis. Due to the presence of numerous null data points in the Ghanaian dataset, it is preprocessed using a cubic spline method. Our model can effectively model two strains. To model the main strains within the given period, strain 1 ($I_1$) consists of the original COVID-19 virus and the Delta variant, while strain 2 ($I_2$) is composed of the Omicron variant. Although the

**Table 2. Estimation of parameters of the model.**

| Parameter | Values | Description | Reference |
|---|---|---|---|
| N | 33,773,888 | The population of Ghana | [47] |
| $\Lambda$ | 28.452 | Daily average of the number of births from 2019–2023 | [48] |
| $\beta_1, \beta_2$ | 0.70, 1.164 | Transmission rate for strain 1, 2 | Fitted |
| $\sigma_1, \sigma_2$ | 1/10, 1/8 | Reciprocal of incubation period for strain 1, 2 | Assumed |
| $\alpha_1, \alpha_2$ | 1/4, 1/4 | Detection rate of infected individuals for strain 1, strain 2 | Assumed |
| $\gamma_1, \gamma_2$ | 1/14, 1/20 | Recovery rates for individuals in $I_1, I_2$ | Assumed |
| $\theta$ | 1/7 | Recovery rate for individuals in $Q$ | [49] |
| $\mu$ | 0.007087 | Natural death rate | [48] |
| $\delta$ | 0.0085 | COVID-19 induced death $\left(\dfrac{\text{deaths } (1,461)}{\text{confirmed cases } (171,018)}\right)$ | [4] |
| $\upsilon$ | 1/90 | Rate at which recovered returns to susceptible | Assumed |

Table contains the values and detailed descriptions of the parameters used in the model and references to them.

exact number of each strain is unknown, the percentage of each strain is calculated by referring to various press releases [46].

We use the preprocessed data to estimate the transmission rate of each strain. The initial conditions are taken on the date of the outbreak. There was only the original virus and no other variants on the first day of the outbreak. The first two COVID-19 cases were reported in Ghana on 12 March 2020 and the total population of Ghana was 33, 773, 888 [3]. It is assumed that there were no recoveries, but rather 8 exposed individuals as the first two confirmed cases during the initial stages. Therefore, the initial values for Ghana are given by $S(0) = 33, 773, 888$, $E_1(0) = 8$, $E_2(0) = 0$, $I_1(0) = 2$, $I_2(0) = 0$, $Q(0) = 0$, $R(0) = 0$. The mutant virus ($I_2$) was first recorded in Ghana on 21 November 2021 [46]. Since we know the number of infected individuals for each strain in the system of ordinary differential equations within our simulation time, we can estimate inversely the parameters. The estimation is done by using the least squares method which consists of minimizing the sum of the squared differences between each observed daily cases data point and the corresponding daily cases point obtained from the model. The values of the parameters used and the values of the estimated parameters are shown in Table 2.

Now we substitute the fitted transmission rate $\beta_1$ and $\beta_2$ into Eq (9) to obtain the basic reproduction number for each strain. The result has been estimated to be $R_0^1 = 1.9396$ and $R_0^2 = 3.4905$, respectively. The preprocessed real data and fitting results are shown in Fig 2.

## Stability of equilibrium

We demonstrated the theoretical stability of our model in the previous section. In this subsection, we verify the stability of the three equilibrium points numerically through simulations. Although the two strains did not actually exist at the time of the outbreak, we assume that the initial infection of both strains occurred at the same time for theoretical confirmation. Also, numerical simulations are undertaken to confirm the theoretical results using the parameters in Tables 2 and 3 except $\beta_1$ and $\beta_2$. We examine three cases for stability and the code is written in MATLAB. Three stability cases are examined:

- Case 1: Disease-free equilibrium. (The simulation results are shown in Fig 3(a))

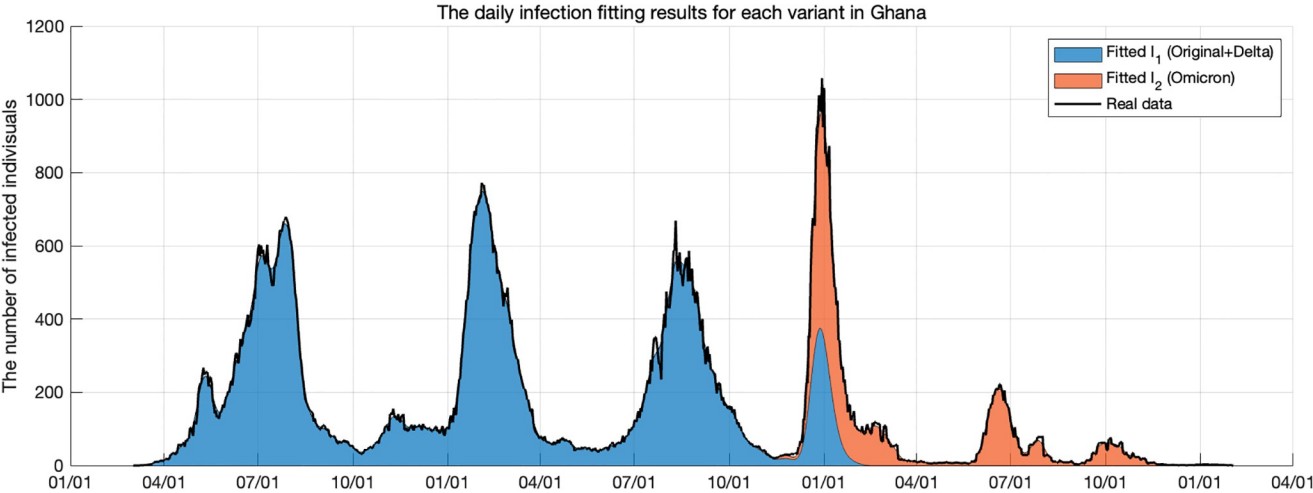

**Fig 2. This figure shows the actual number of daily COVID-19 cases (black lines) and fitting results (blue and red areas) of each strain after preprocessing.** The blue area shows the fitting results of the original virus and Delta variant, and the red area shows the fitting results of the Omicron variant.

- Case 2: Outbreak of strain 1; Original COVID-19 Virus and Delta Variant. (The simulation results are shown in Fig 3(b))

- Case 3: Outbreak of strain 2; Omicron Variant. (The simulation results are shown in Fig 3(c))

Fig 3(a) shows the dynamics of the infections of the multi-strain model at the stability of disease-free equilibrium. To simulate a disease-free state, the basic reproduction number must be less than 1. Therefore, we take $\beta_1$ and $\beta_2$ as 0.1, setting the basic reproduction number of each strain to less than 1 ($R_0^1 = 0.2271$ and $R_0^2 = 0.2999$). The transmission rate used for each case is shown in Table 3. From Fig 3(a), the curves of $E_1, E_2, I_1, I_2, Q$ dropped to zero, this is in line with the theoretical result in Theorem 4. Fig 3(b) describes the equilibrium of strain 1 and could be observed that as the first strain of the disease exists the other strain dies out. The basic reproduction number of strain 1 is greater than 1 ($R_0^1 = 1.6625 \geq 1$) while the basic reproduction number of strain 2 is less than 1 ($R_0^2 = 0.2999 \leq 1$). Similarly, Fig 3(c) describes the equilibrium of strain 2 and could be observed that as the second strain of the disease increases, the other strain dies out. The basic reproduction number of strain 2 is greater than 1 ($R_0^2 = 1.7992 \geq 1$) while the basic reproduction number of strain 1 is less than 1

**Table 3. Transmission rates for each case.**

| Transmission rates | Case 1 | Case 2 | Case 3 |
|---|---|---|---|
| $\beta_1$ ($R_0^1$) | 0.1 (0.2271) | 0.6 (1.6625) | 0.1 (0.2271) |
| $\beta_2$ ($R_0^2$) | 0.1 (0.2999) | 0.1 (0.2999) | 0.6 (1.7992) |

Table shows the transmission rates for implementing the three stability of equilibrium. First, a disease-free equilibrium is assumed where both $R_0^1$ and $R_0^2$ are less than 1. To represent the outbreak of strain 1 and strain 2, it is assumed that $R_0^1$ and $R_0^2$ are greater than 1, respectively.

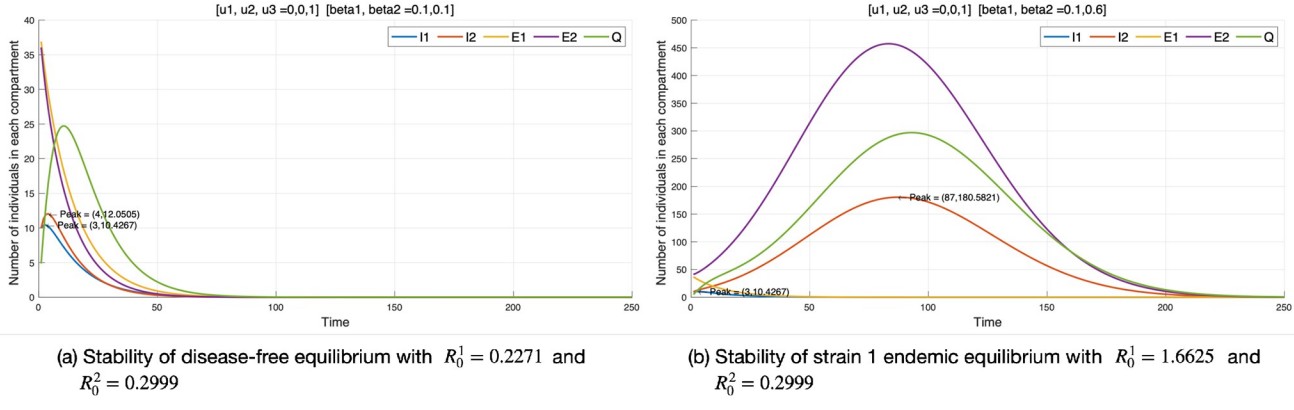

(a) Stability of disease-free equilibrium with $R_0^1 = 0.2271$ and $R_0^2 = 0.2999$

(b) Stability of strain 1 endemic equilibrium with $R_0^1 = 1.6625$ and $R_0^2 = 0.2999$

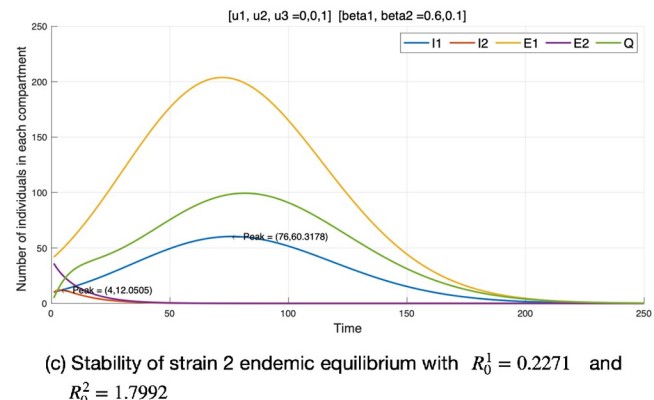

(c) Stability of strain 2 endemic equilibrium with $R_0^1 = 0.2271$ and $R_0^2 = 1.7992$

**Fig 3. Stability of the three equilibria using the cases in Table 3.**

($R_0^1 = 0.2271 \leq 1$). The numerical simulation results are consistent with the theoretical results carried out.

### Control strategies assessment

In this section, we consider the control strategies (17) utilized to reduce COVID-19 in Ghana. We observe the three control strategies namely the social distancing ($u_1$), vaccination ($u_2$), and testing-treatment of COVID-19 patients ($u_3$) and its effect on the spread of COVID-19. We consider the effects of the combination of two control strategies on COVID-19 and its effect on each compartment. The control strategies are also viewed at different levels (mild, average, and strict).

**Control strategy 1.** With the first control (social distancing), we consider $u_1 \neq 0$ while other controls ($u_2$, $u_3$) are set to zero. Additionally, the effect of this control is being viewed at a mild level ($u_1 = 0.1$), average level ($u_1 = 0.5$), and strict level ($u_1 = 0.8$). From Fig 4, we observe a significant reduction in the number of infected individuals at different control levels. The reduction is as follows: from millions ($10^6$) at a mild level ($u_1 = 0.1$), to ten thousand ($10^4$) at an average level ($u_1 = 0.5$), and tens ($10s$) at a strict level ($u_1 = 0.8$), indicating that stricter measures help delay the peak of the disease. This could be seen in the efforts of the Government of Ghana as the level of social distancing became stricter. Public gatherings, which include religious activities and funerals, were banned, public transportation was operated with

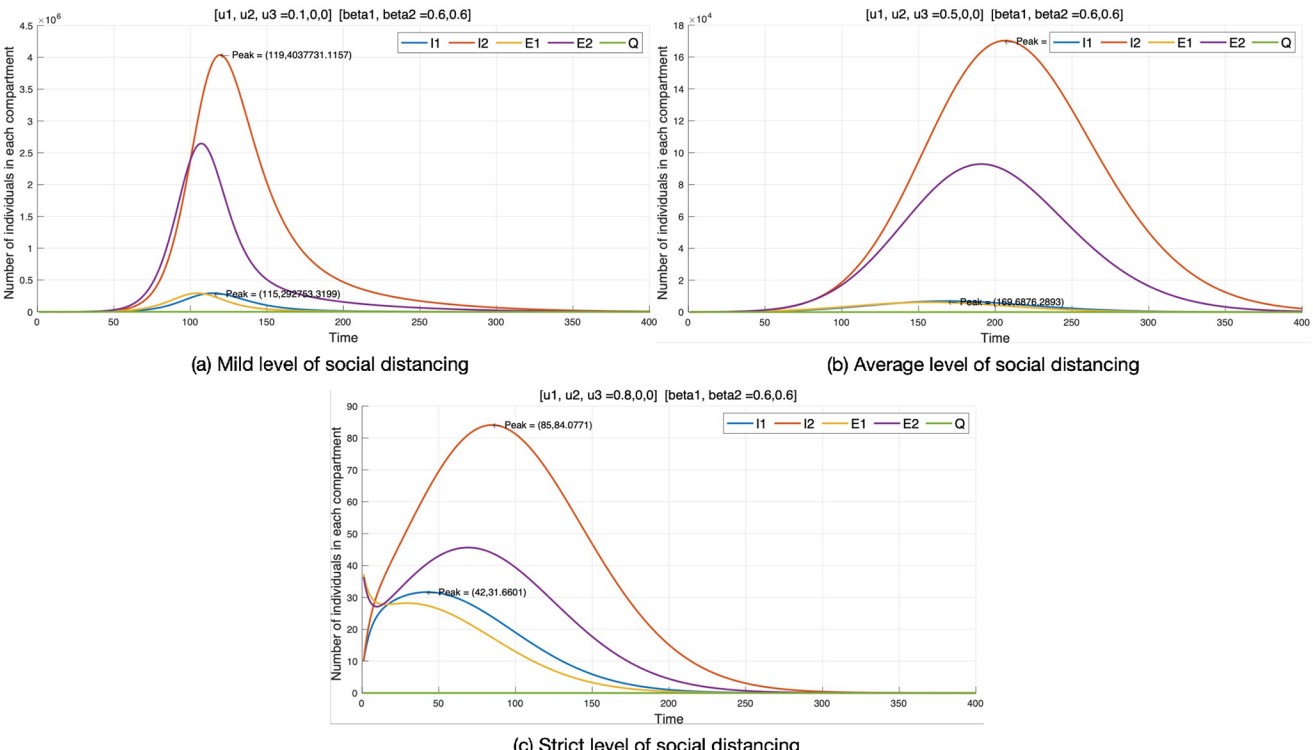

**Fig 4. Control strategy 1.**

a minimal number of passengers, and individuals who did not wear face masks were penalized [50]. The Ghanaian borders, both air and land, were closed to reduce COVID-19 infected cases.

Additionally, we observe the effect of the control at mild, average, and strict levels on each compartment ($I_1$, $I_2$, $E_1$, $E_2$, $Q$). The effect of the control strategy at different levels is considered on $I_1$. At the mild level, 300,000 individuals are infected, taking 115 days to reach their peak, the highest point after the outbreak. At the average level with $u_1$ implemented, 10,000 individuals are infected with the original virus, taking 169 days to reach their peak. At the strict level ($u_1 = 0.8$), there are 30 individuals infected, taking 42 days to reach its peak. Regarding the mutant virus ($I_2$), at the mild level of $u_2$, 4 million individuals are infected, taking 119 days to reach their peak. At the average level with $u_2$ implemented, there are 170,000 individuals infected, taking 206 days to reach its peak. At the strict level ($u_2 = 0.8$), there are 85 individuals infected, taking 85 days to reach its peak. The analysis of control 1 (social distancing) is summarized in Table 4.

In conclusion, the number of infected individuals with the mutant virus ($I_2$) is significantly higher than that of the original virus ($I_1$) at all levels. Additionally, the stricter the control, the fewer the number of infected and exposed individuals to the virus. The peak increases from the mild to average level, indicating there would be enough time to prepare before it reaches the peak. In other words, the infection curve flattened [39]. The effect is to delay the peak of infected individuals and postpone the eradication of the infectious disease. However, there is some reduction when the control strategy is implemented at a strict level. This means that strong social distancing measures go beyond flattening the infection curve and lead to a

**Table 4. Control strategy 1.**

| Social distancing | | I1 | I2 | E1 | E2 |
|---|---|---|---|---|---|
| Mild level ($u_1 = 0.1$) | Population | $3 \times 10^6$ | $4 \times 10^7$ | $2 \times 10^6$ | $2.7 \times 10^7$ |
| Average level ($u_1 = 0.5$) | | 10,000 | $1.7 \times 10^6$ | 10,000 | 90,000 |
| Strict level ($u_1 = 0.8$) | | 30 | 85 | 29 | 46 |
| Mild level ($u_1 = 0.1$) | Peak (days) | 115 | 119 | 110 | 130 |
| Average level ($u_1 = 0.5$) | | 169 | 206 | 170 | 190 |
| Strict level ($u_1 = 0.8$) | | 42 | 85 | 40 | 90 |

Table notes the number of individuals at the peak of each compartment of optimal control strategy 1 and the period until the peak occurs.

significant reduction in the number of infected individuals, thus advancing the endpoint of the endemic. Therefore, it is best when $u_1 = 0.8$ (strict level); however, in the real world it would be difficult. Hence, it is recommended to use $u_1$ as strongly as possible.

**Control strategy 2.** Here, we are examining the second control strategy (vaccination) with $u_2 \neq 0$, while the other controls ($u_1$, $u_3$) are set to zero. Fig 5 presents the experimental results. As a result, we find that there are more individuals infected with $I_2$ than those infected with $I_1$. However, we observe that population vaccination leads to a greater reduction in diseases. The vaccination control strategy shows a minimum number of individuals in each compartment. Regarding $I_1$, vaccination implemented at a mild level shows that 70 individuals are infected with the virus, with its peak at 29 days. At an average level ($u_2 = 0.5$), there are 26

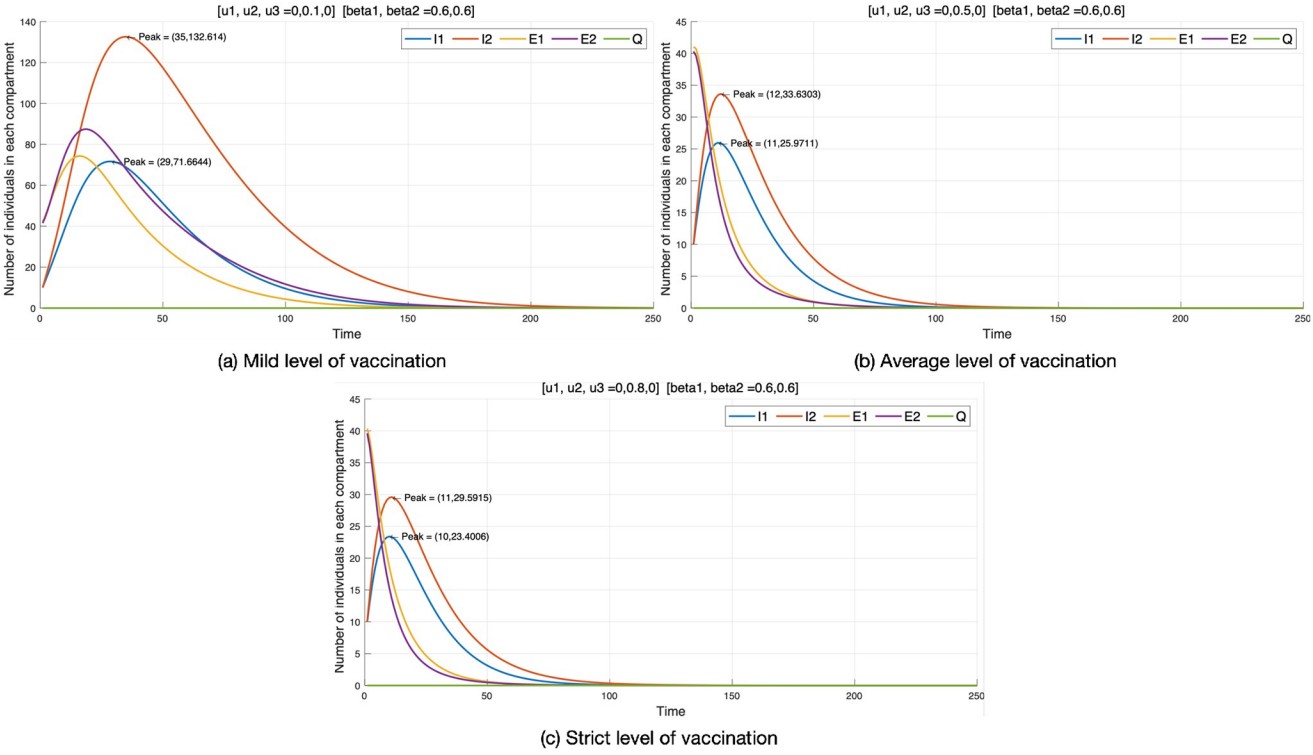

**Fig 5. Control strategy 2.**

**Table 5. Control strategy 2.**

| Vaccination | | $I_1$ | $I_2$ | $E_1$ | $E_2$ |
|---|---|---|---|---|---|
| Mild level ($u_2 = 0.1$) | Population | 70 | 130 | 70 | 85 |
| Average level ($u_2 = 0.5$) | | 26 | 33 | Terminates | Terminates |
| Strict level ($u_2 = 0.8$) | | 23 | 29 | | |
| Mild level ($u_2 = 0.1$) | Peak (days) | 29 | 35 | 10 | 40 |
| Average level ($u_2 = 0.5$) | | 11 | 12 | No outbreak | No outbreak |
| Strict level ($u_2 = 0.8$) | | 10 | 11 | | |

Table notes the number of individuals at the peak of each compartment of optimal control strategy 2 and the period until the peak occurs. 'No outbreak' signifies compartments without significant outbreaks. 'Terminate' indicates compartments where the outbreak terminates without reaching a significant peak.

individuals infected, with the peak occurring in 11 days. At a strict level, there is not much difference compared to the mild level, with 23 individuals infected and a peak occurring in 10 days.

With individuals infected with the mutant virus ($I_2$) when $u_2 = 0.1$, we observe that there are 130 individuals infected, with its peak at 35 days. At an average level, there is a drastic reduction in the number of individuals infected (33 individuals), with its peak at 12 days. There is not much difference when $u_2$ is implemented at a strict level compared to a mild level, with 29 individuals getting infected and its peak at 11 days. In conclusion, the vaccination strategy is found to be highly effective compared to Control strategy 1. The analysis of the vaccination strategy (Control strategy 2) is summarized in Table 5.

**Control strategy 3.** In control strategy 3, only testing-treatment of COVID-19 patients is applied. We consider the effect on the individual compartments at three levels: mild, average, and strict. From Fig 6, at different control levels, the number of individuals reduces dramatically from millions ($10^6$) for a mild level $u_1 = 0.1$, ten thousand ($10^4$) for an average level $u_1 = 0.5$, and a thousand ($10^3$) for a strict level $u_1 = 0.8$, respectively. This indicates a reduction in the number of individuals as the measures become stricter, helping to delay the peak of the disease.

At 10% compliance, infected individuals usually do not disclose their status. Therefore, the number of patients in the quarantine compartment is very small. The low number of quarantined individuals suggests that the actual number of infected individuals may be higher than reported by the government. In fact, governments use the number of individuals revealed by COVID-19 tests to estimate the number of infected individuals [45]. Since we assume quarantined individuals do not transmit the disease, 10% compliance makes it difficult to determine the number of exposed individuals accurately. Therefore, there are more infected individuals ($I_1$, $I_2$) than exposed individuals ($E_1$, $E_2$) in this case. In other words, infected individuals in Fig 6(a) may include individuals whom the government does not identify. On the other hand, we can determine a more accurate number of exposed individuals and observe a reduction in infected individuals when patients are 50% compliant with testing-treatment. Additionally, infection rates significantly decrease at 80% compliance, and exposed individuals become more accurately determined.

The exposed compartment $E_1$ at a mild level of $u_3$ shows that 300,000 individuals are exposed to the disease, with 10,000 individuals at an average level, and only a few individuals at a strict level of testing-treatment (600 individuals). Similarly, $E_2$ shows a significant reduction in the number of individuals exposed to the mutant virus as $u_3$ is implemented at various levels (mild, average, and strict).

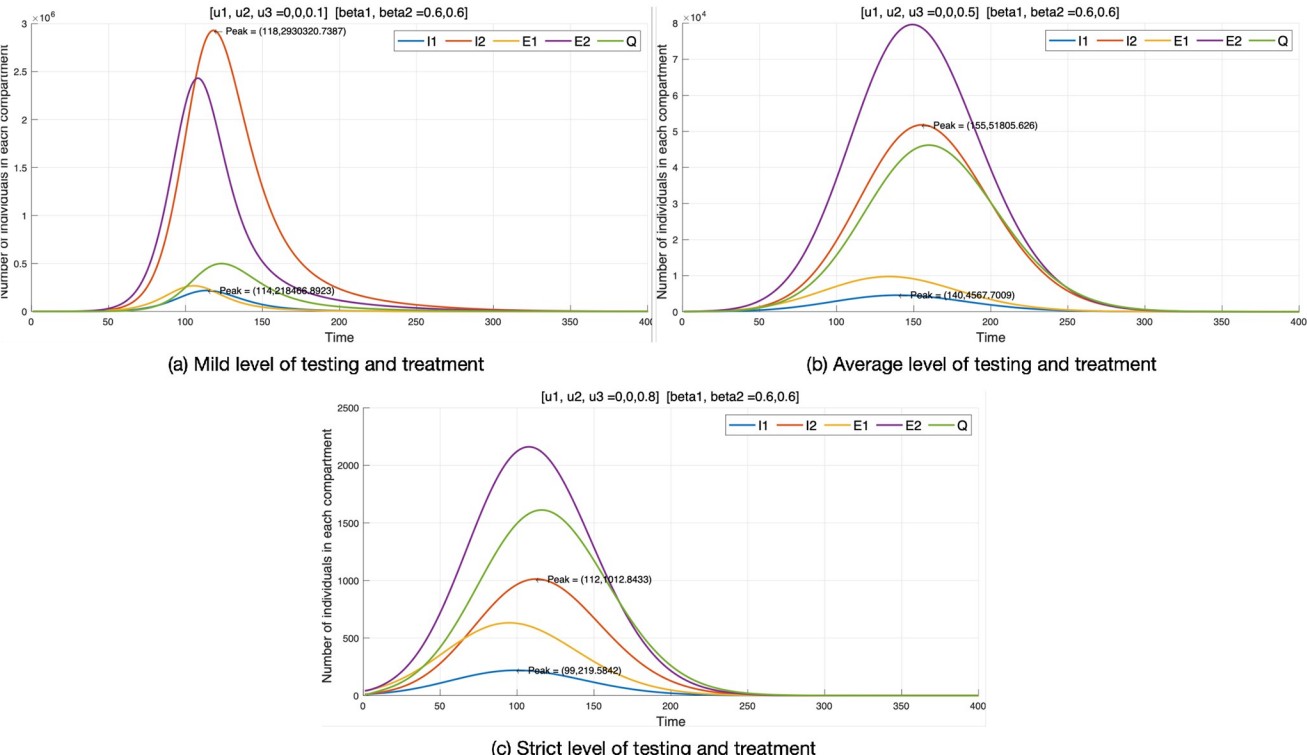

(a) Mild level of testing and treatment

(b) Average level of testing and treatment

(c) Strict level of testing and treatment

**Fig 6. Control strategy 3.**

Overall, the results of control strategy 3 show similar trends to those of control strategy 1. Intensifying testing-treatment from mild to average levels flattens the curve, and strict levels of testing-treatment accelerate the end of the disease. What distinguishes this strategy from others is its ability to determine the number of individuals exposed according to the level of testing-treatment. The fact that the number of exposed individuals is less than the number of infected individuals suggests that there are a significant number of infected individuals that the government is not aware of. This underscores the importance of high-level testing-treatment of COVID-19 in establishing appropriate response strategies.

**Combination of two control strategies.** In the case of control strategy 2, since it has a strong effect on reducing the number of infected individuals, it still shows a strong effect when combined with other control strategies. Therefore, we mainly present the result of combining two control strategies 1 and 3 in this subsection. Additionally, the entire experimental results are attached in S2 Appendix.

In Fig 7, we consider control strategy 1 (social distancing) and control strategy 3 (testing-treatment of COVID-19 patients) at a mild, average, and strict level. We set up three experiments:

1. Comparison of both control strategies 1 and 3 at the mild level and both at the strict level (Fig 7(a) and 7(b)).

2. Comparison of one control strategy at the mild level and the other at the average level (Fig 7(c) and 7(d)).

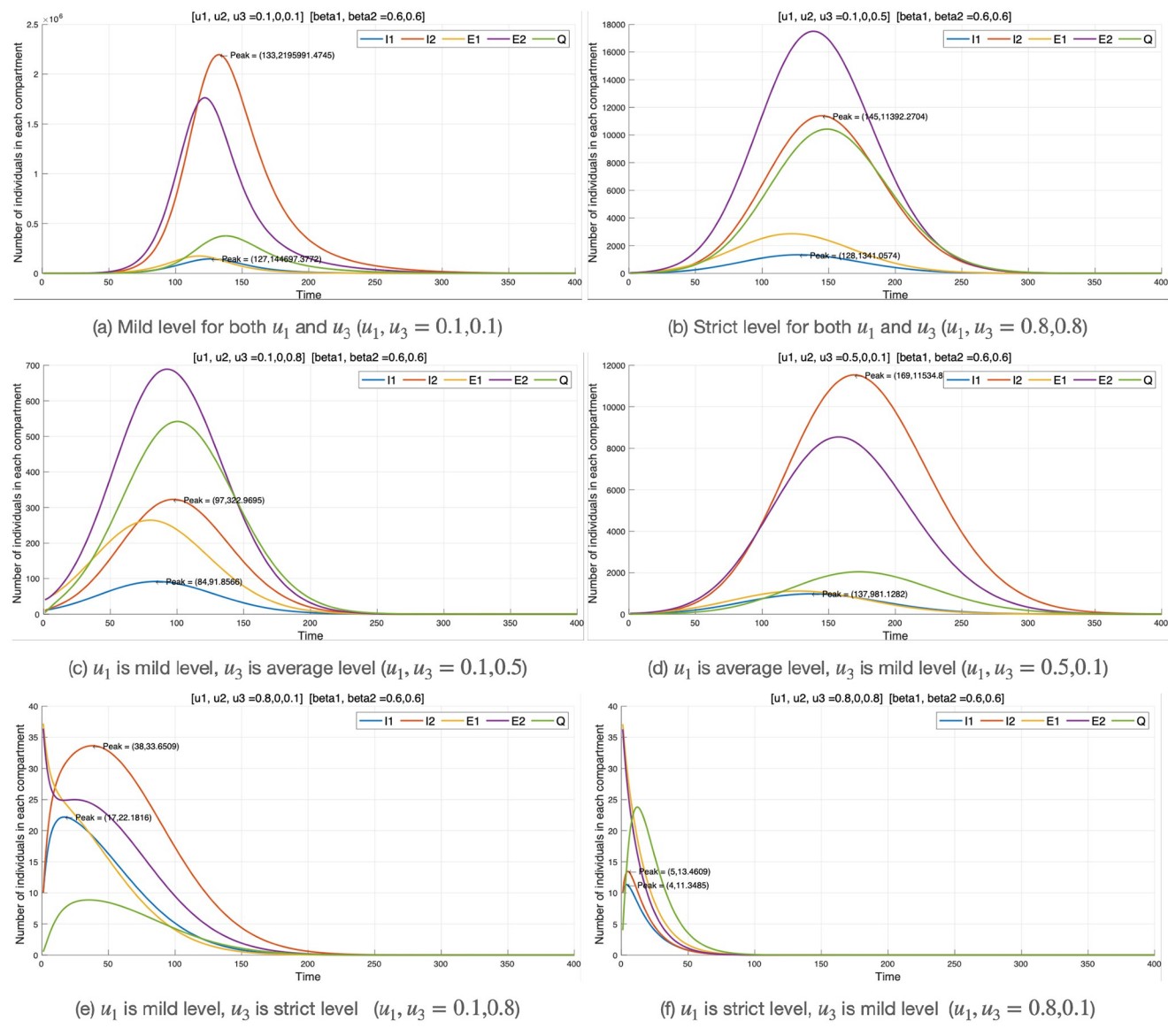

**Fig 7. Control strategy 1 and 3.**

3. Comparison of one control strategy at the mild level and the other at the strict level (Fig 7 (e) and 7(f)).

The results of the first experiment can be found in Fig 7(a) and 7(b). It is observed that implementing $u_1$ and $u_3$ at a strict level leads to a significant reduction in the disease compared to being at a mild level. Table 6 presents the distribution of individuals in each compartment as a result of the experiment.

We assess the effect of $u_1$ and $u_3$ on individual compartments. With respect to the implementation of $u_1$ and $u_3$, 144,697 individuals are infected with the original and Delta variant ($I_1$) at a mild level, peaking at 127 days. At a strict level, only 11 individuals are infected, peaking at 4 days for both mild and strict levels, respectively. For $I_2$, 2,195,991 individuals are

**Table 6. Control 1 & 3 strategy.**

| Social distancing and testing-treatment | | $I_1$ | $I_2$ | $E_1$ | $E_2$ | $Q$ |
|---|---|---|---|---|---|---|
| $u_1, u_3 = 0.1, 0.1$ | Population | 144,697 | 2,195,991 | 62 | 70 | 61 |
| | Peaks (days) | 127 | 133 | 27 | 27 | 27 |
| $u_1, u_3 = 0.8, 0.8$ | Population | 11 | 13 | No outbreak | No outbreak | 61 |
| | Peaks (days) | 4 | 5 | | | 27 |
| $u_1, u_3 = 0.1, 0.5$ | Population | 1,341 | 11,392 | 2,500 | 17,000 | 10,000 |
| | Peaks (days) | 128 | 145 | 125 | 142 | 150 |
| $u_1, u_3 = 0.5, 0.1$ | Population | 981 | 11,534 | 1000 | 8200 | 2,000 |
| | Peaks (days) | 137 | 169 | 120 | 155 | 175 |
| $u_1, u_3 = 0.1, 0.8$ | Population | 98 | 310 | 200 | 695 | 550 |
| | Peaks (days) | 84 | 97 | 60 | 98 | 100 |
| $u_1, u_3 = 0.8, 0.1$ | Population | 22 | 33 | No outbreak | No outbreak | No outbreak |
| | Peaks (days) | 17 | 38 | | | |

Table notes the number of individuals at the peak of each compartment of the mixed strategy of optimal control strategies 1 and 3 and the period until the peak occurred. 'No outbreak' signifies compartments without significant outbreaks.

infected at a mild level, peaking at 133 days, and only 13 individuals are infected at a strict level, peaking at 5 days. Additionally, we consider exposed individuals ($E_1$, $E_2$) at both levels. Many individuals are exposed to the COVID-19 virus at a mild level, while there are no outbreaks at a strict level. In the quarantine compartment ($Q$), few individuals are quarantined at a mild level, whereas at the strict level, individuals exposed and infected with the disease go into quarantine to prevent further spread. The combination of $u_1$ and $u_3$ at a mild level is the worst combination of control strategies to be implemented, as many individuals are exposed to the virus and become infected without adhering to social distancing or testing-treatment for COVID-19.

Likewise, we compare the effects of $u_1$ and $u_3$ at mild and average levels, respectively (Fig 7 (c)), and vice versa, with $u_1$ and $u_3$ at average and mild levels, respectively (Fig 7(d)). Regarding $I_1$, when $u_1$ is at a mild level and $u_3$ is at an average level, there are 1,341 individuals infected with the virus, peaking at 128 days compared to when $u_1$ is at an average level and $u_3$ is at a mild level with 981 individuals infected and peaking at 137 days. Similarly, for $I_2$, when $u_1$ is at a mild level and $u_3$ is at an average level, there are 11,392 individuals infected with the virus, peaking at 145 days compared to when $u_1$ is at an average level and $u_3$ is at a mild level with 11,534 individuals infected and peaking at 169 days. These results indicate that the two strategies produce similar numbers of infected individuals. However, from the perspective of the exposed individuals, the two results show a significant difference. Comparing exposed individuals from Fig 7(c) and 7(d), it is estimated that there are more exposed individuals in Fig 7(c). As mentioned in control strategy 3, this means that even with the same level of infection rate, better quarantine measures will give the government a more accurate picture of the number of infected individuals. Therefore, if the government has the option to choose either social distancing or COVID-19 testing at an average level, it would be advisable to choose COVID-19 testing.

Finally, in the mild and strict combination of the control strategies, Fig 7(f) shows a significant reduction in the number of exposed and infected individuals when $u_1 = 0.8$ and $u_3 = 0.1$ compared to Fig 7(e) when $u_1 = 0.1$ and $u_3 = 0.8$. This means that, unlike the results of Experiment 2, strict social distancing is more effective than a strict testing-treatment strategy. A

significant finding emerges when comparing the number of infected individuals at the average level ($u_1 = 0.5$ or $u_3 = 0.5$) in Table 6 with the average level ($u_1 = 0.5$) in Table 4. These results demonstrate that implementing both strategies simultaneously leads to a 1/10 reduction in the number of infected individuals, even if one of the strategies is implemented at a mild level.

In summary, implementing the two strategies together greatly contributes to reducing the number of infected individuals. Additionally, the testing-treatment strategy is more effective if the government has the option to choose the average level between the two strategies ($u_1$, $u_3$), and social distancing is more effective if the government can choose the strict level.

## Discussion and conclusions

COVID-19 has been recorded as one of the deadliest pandemics in human history, infecting more than 690 million people around the world and causing more than 6.9 million deaths [45]. It is well known from several studies [21, 27, 28, 39] that implementing strong control strategies with quick decisions in the early stages of an outbreak prevents the explosive increase of the infected individuals. However, excessive control strategies put people into fatigue and limit their outdoor activities. Therefore, choosing an effective control strategy has become very important. Many organizations and researchers have used mathematical modeling to estimate the efficiency of newly applied control strategies. On the other hand, sometimes mathematical models can be vulnerable to the sudden emergence of a mutant virus, which is more contagious than an original virus, since they were specific to the region and circumstances at the time.

In this study, we propose a mathematical model considering multiple strains, and simulate optimal control by selecting common strategies implemented in many countries, such as social distancing, vaccination, testing-treatment. The model was established in terms of the positivity and boundedness of the solution and the existence of three equilibrium points (disease-free equilibrium, strain 1 endemic equilibrium, and strain 2 endemic equilibrium). We derived the basic reproduction numbers $R_0^1$ and $R_0^2$ for each strain based on the mathematical model and established the local stability of the solutions of this model. Furthermore, numerical simulations were conducted to validate the theoretical proof of the equilibrium model. Fitting the data of the infected in Ghana into the model, we found that the basic reproduction number of the Omicron variant ($R_0^2 = 3.4905$) in Ghana is measured to be 1.8 times higher than that of the original and Delta variant ($R_0^1 = 1.9396$).

We derived optimal control models for three control strategies (social distancing, vaccination, and testing-treatment) using the Pontryagin's Maximum Principle. These three control strategies were once again divided into three categories: mild (0.1), moderate (0.5), and strict (0.8) level, and we performed a rigorous analysis by simulating all combinations of possible control strategies. The analysis was conducted by comparing the peak number of individuals in each compartment with the date on which the peak occurred to take into account the impact on individual compartments. As a result, we found that the vaccination strategy was more efficient than the other two control strategies. However, it was very difficult to develop vaccines early in the outbreak of the COVID-19. Assuming the same level of social distancing and testing-treatment strategy, the effectiveness of both strategies was similar. However, we observed that if the testing-treatment strategy was not implemented well, there would be more confirmed individuals than the estimated exposed individuals. This result suggests that there are many infected individuals, which were caused by the assumption 4 (The infected groups ($I_i$) for each strain include both symptomatic and asymptomatic individuals, where $i = 1, 2$). Therefore, when the government weakens its control strategies, it must make sure that the testing-treatment strategy is being implemented properly. If the testing-treatment strategy is not

implemented sufficiently, reducing the intensity of the control strategy can lead to a sharp increase in the number of infected individuals.

Finally, we simulated situations where two strategies were implemented simultaneously. In these simulations, similar to the scenario where only one control strategy was implemented, we obtained results indicating that if the testing-treatment strategy is not effectively implemented, more confirmed individuals occur than individuals estimated to be exposed. Furthermore, when two control strategies are implemented simultaneously, even if one control strategy is implemented at a mild level, the number of infected individuals decreases to 1/10 compared to when only one strategy is implemented. This result suggests the effectiveness of using a combination of two control strategies in reducing the number of infected individuals.

It is possible to adjust the intensity of each optimal strategy over time using our model. However, we focused on comparing the impact of varying the initial intensity of strategy implementation and discussing which combinations of strategies are more efficient. Therefore, we needed to fix the levels of each strategy to compare their effectiveness in this study. We leave an optimization of the implementation intensity of strategies over time as a future study.

Our model is a novel model as it is a multi-strain model that has not been widely studied in Ghana. Additionally, while most studies with optimal control strategies look at each strategy individually in their COVID-19 models, our model also considered various combinations of control strategies. Furthermore we find that implementing control strategies reduces the spread of COVID-19 and helps the government of Ghana and the Ministry of Health (MOH) with strategies to control the disease.

## Supporting information

**S1 Appendix. Computation of basic reproduction number and proof of existence of the equilibrium points.**
(PDF)

**S2 Appendix. Combination of two control strategies.** This appendix contains all combinations of the two control strategies. Therefore, it includes combinations of 1 and 2, 2 and 3, and 1 and 3 and their results analysis.
(PDF)

**S1 File. Data and code.** This material includes the data from Ghana we used and the code written in MATLAB to obtain data fitting results and optimal control results.
(ZIP)

## Author Contributions

**Conceptualization:** Young Rock Kim, Joy Nana Okogun-Odompley.

**Data curation:** Youngho Min, Joy Nana Okogun-Odompley.

**Formal analysis:** Young Rock Kim, Youngho Min, Joy Nana Okogun-Odompley.

**Funding acquisition:** Young Rock Kim.

**Investigation:** Young Rock Kim, Youngho Min, Joy Nana Okogun-Odompley.

**Methodology:** Young Rock Kim, Youngho Min, Joy Nana Okogun-Odompley.

**Project administration:** Young Rock Kim.

**Resources:** Young Rock Kim.

**Software:** Youngho Min.

**Supervision:** Young Rock Kim.

**Validation:** Youngho Min.

**Visualization:** Youngho Min.

**Writing – original draft:** Youngho Min, Joy Nana Okogun-Odompley.

**Writing – review & editing:** Young Rock Kim, Youngho Min.

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
