## [Decision Letter · Decision Letter 0]

26 Dec 2023

PONE-D-23-27070A Multi-strain Mathematical Modeling of COVID-19 with Optimal Control in Ghana: Assessing the Impact of Governmental and Private Interventions such as Social Distancing, Vaccination, Testing and TreatmentPLOS ONE

Dear Dr. Min,

Thank you for submitting your manuscript to PLOS ONE. After careful consideration, we feel that it has merit but does not fully meet PLOS ONE’s publication criteria as it currently stands. Therefore, we invite you to submit a revised version of the manuscript that addresses the points raised during the review process.

We look forward to receiving your revised manuscript.

Kind regards,

Rehana Naz

Academic Editor

PLOS ONE

Journal Requirements:

2. Please note that PLOS ONE has specific guidelines on code sharing for submissions in which author-generated code underpins the findings in the manuscript. In these cases, all author-generated code must be made available without restrictions upon publication of the work. 

Please review our guidelines at https://journals.plos.org/plosone/s/materials-and-software-sharing#loc-sharing-code and ensure that your code is shared in a way that follows best practice and facilitates reproducibility and reuse.

"This work was supported by the Hankuk University of Foreign Studies Research Fund. Young Rock Kim and Youngho Min were supported by the National Research Foundation of Korea (NRF) grant funded by the Korea government (MSIT) (No. 2021 R1A2C1011467). The funders had no role in study design, data collection and analysis, decision to publish, or preparation of the manuscript."

We note that one or more of the authors is affiliated with the funding organization, indicating the funder may have had some role in the design, data collection, analysis or preparation of your manuscript for publication; in other words, the funder played an indirect role through the participation of the co-authors. If the funding organization did not play a role in the study design, data collection and analysis, decision to publish, or preparation of the manuscript and only provided financial support in the form of authors' salaries and/or research materials, please do the following:

(1) Review your statements relating to the author contributions, and ensure you have specifically and accurately indicated the role(s) that these authors had in your study. These amendments should be made in the online form.

(2) Confirm in your cover letter that you agree with the following statement, and we will change the online submission form on your behalf: 

Reviewers' comments:

Reviewer's Responses to Questions

**Comments to the Author**

1. Is the manuscript technically sound, and do the data support the conclusions?

Reviewer #1: Partly

Reviewer #2: Partly

2. Has the statistical analysis been performed appropriately and rigorously? 

Reviewer #1: I Don't Know

Reviewer #2: I Don't Know

3. Have the authors made all data underlying the findings in their manuscript fully available?

Reviewer #1: No

Reviewer #2: Yes

4. Is the manuscript presented in an intelligible fashion and written in standard English?

Reviewer #1: No

Reviewer #2: No

5. Review Comments to the Author

Reviewer #1: The results listed in the paper in the form of formulas, figures, and analysis seems true and correct. The paper is well written and it is written in a truly sporty manner. English is generally good, I think it needs to be polished further and some typos need to be revised. Further punctuation marks should be checking through the paper, especially after the equations and at the end of the statements.

* Remark, comments, and questions:

----- Title of paper is not clear. Try to clear meaning of the title.

------ The abstract is a little thin and does not quite convey the vibrancy of the findings and the depth of the main conclusions. The authors should please extend this somewhat for a better first impression.

------ The manuscript lacks motivation. Author needs to include the motivation of the study.

------Authors should write keywords in professional way.

----- There is already an abundance of modeling studies on COVID-19, vaccinations, and the months or years to come. However, apart from Ferguson's (now classic) work, Moore and Giordano, very little is said about similar modeling works. This is an issue for three reasons. First, the intended audience for such pieces is made of policy-makers and the general public: they are already facing an abundance of (occasionally conflicting) findings from models. If there is no attempt to contextualize the findings from this piece among others, then we're more likely to be adding noise to a crowded space, instead of providing valuable guidance. Second, several of the modeling assumptions made here may be in line with other pieces (which may provide some strength to the methods) or may be rather unique (which may need more discussion). Finally, as a piece of scientific literature, the contributions should be situated based on what already exists. In sum, the authors should explain how each of their assumptions and modeling choices compares to the literature; how their findings compare to the literature; and hence what is their specific contribution. Related models include, but are not limited to:

--https://doi.org/10.1016/j.rinp.2021.104285

--https://doi.org/10.1007/s12190-021-01507-y

--https://doi.org/10.1016/j.chaos.2020.110173

--https://doi.org/10.1063/5.0016240

--https://doi.org/10.1016/j.chaos.2020.110049

--arXiv:2005.06286v2

--https://www.researchsquare.com/article/rs-872671/v1

-- https://doi.org/10.1140/epjp/s13360-021-01997-6

-- https://doi.org/10.1140/epjp/s13360-022-02347-w

-- https://doi.org/10.1007/s11071-022-07235-7

----Is there any experimental data to validate the mathematical model ? The authors at least describe the basic reproduction number R_0 and its impact on covid-19 pandemic. The basic reproduction number R_0 is one of the most crucial quantities in infectious diseases, as R_0 measures how contagious a disease is. In this context the authors include the reference "Mathematical analysis of the global dynamics of a HTLV-I infection model, considering the role of cytotoxic T-lymphocytes, Math. Comput. Simul. 180(2021) 354-378", “Dynamics of an HTLV-I infection model with delayed CTLs immune response, Appl Math Comput 2022”

------Authors should insert all figures in appropriate places.

------Conclusion should be written in a more clear way. So try to short it and write in a professional way.

------Analysis is missing in paper so add it.

------Authors should improve the English of paper.

------Authors should correct grammatical error at few stage of paper.

-------Presentation of paper should be improved.

-------Try to reduce similarity of work.

-------References list are not appropriate.

Reviewer #2: I have two major concerns regarding the study.

1. A very important pathway in the transmission of COVID-19 was the asymptomatic population, this is missing in the model. This makes the model rather unspecific, and in particular may not capture the dynamics of COVID-19 very well.

2. The novelty of the study is not clear, most of the work presented already exists in the literature. Perhaps estimation of the contact rates (and hence R_0) is an addition to the existing literature, but this section is not very clear, at the very least fitting results (which it seems have been obtained by fitting several waves) need to be presented in detail.

Further, the language really needs to be fixed, some of the headings and many sentences throughout the study are not grammatically correct.

6. PLOS authors have the option to publish the peer review history of their article (what does this mean?). If published, this will include your full peer review and any attached files.

Reviewer #1: No

Reviewer #2: **Yes**

---

## [Author Response · Author response to Decision Letter 0]

9 Feb 2024

We would like to thank the reviewers and editors for the positive feedback and helpful comments on corrections or corrections. Detailed responses to each reviewer's comments are in the attached pdf file "Response on Reviewers".

---

## [Decision Letter · Decision Letter 1]

24 Mar 2024

PONE-D-23-27070R1A mathematical model of COVID-19 with multiple variants of the virus under optimal control in GhanaPLOS ONE

Dear Dr. Min,

Thank you for submitting your manuscript to PLOS ONE. After careful consideration, we feel that it has merit but does not fully meet PLOS ONE’s publication criteria as it currently stands. Therefore, we invite you to submit a revised version of the manuscript that addresses the points raised during the review process.

We look forward to receiving your revised manuscript.

Kind regards,

Rehana Naz

Academic Editor

PLOS ONE

Journal Requirements:

Reviewers' comments:

Reviewer's Responses to Questions

**Comments to the Author**

1. If the authors have adequately addressed your comments raised in a previous round of review and you feel that this manuscript is now acceptable for publication, you may indicate that here to bypass the “Comments to the Author” section, enter your conflict of interest statement in the “Confidential to Editor” section, and submit your "Accept" recommendation.

Reviewer #1: All comments have been addressed

Reviewer #2: (No Response)

Reviewer #3: All comments have been addressed

2. Is the manuscript technically sound, and do the data support the conclusions?

Reviewer #1: Yes

Reviewer #2: Partly

Reviewer #3: Yes

3. Has the statistical analysis been performed appropriately and rigorously? 

Reviewer #1: Yes

Reviewer #2: Yes

Reviewer #3: Yes

4. Have the authors made all data underlying the findings in their manuscript fully available?

Reviewer #1: (No Response)

Reviewer #2: Yes

Reviewer #3: Yes

5. Is the manuscript presented in an intelligible fashion and written in standard English?

Reviewer #1: Yes

Reviewer #2: Yes

Reviewer #3: Yes

6. Review Comments to the Author

Reviewer #1: Authors incorporated all the comments and the manuscript accepted for publication. Now, the manuscript improved a lot.

Reviewer #2: The authors did address some of the points but the main concerns regarding the manuscript remain. Please see attached file for details.

Reviewer #3: I have gone through the responses to reviewers comments and the revised version of the manuscript with marked changes. I can affirm that the authors have asignificantly addressed all the queries and also improve the manuscript. Therefore, it can be accepted for publication in its current form.

7. PLOS authors have the option to publish the peer review history of their article (what does this mean?). If published, this will include your full peer review and any attached files.

Reviewer #1: No

Reviewer #2: **Yes**

Reviewer #3: No

---

## [Author Response · Author response to Decision Letter 1]

23 Apr 2024

Thank you for your feedback. We have addressed the concerns raised by the second reviewer by enhancing Figure 1 to provide a clearer explanation of the model. Additionally, we have clarified the scope of our research in both the Introduction and Discussion sections. For further details, please refer to the attached "Response to Reviewers" document. We have extensively discussed and incorporated your feedback into the manuscript. We kindly request your thoughtful consideration and evaluation of these revisions.

---

## [Decision Letter · Decision Letter 2]

1 May 2024

A mathematical model of COVID-19 with multiple variants of the virus under optimal control in Ghana

PONE-D-23-27070R2

Dear Dr. Min,

We’re pleased to inform you that your manuscript has been judged scientifically suitable for publication and will be formally accepted for publication once it meets all outstanding technical requirements.

Kind regards,

Rehana Naz

Academic Editor

PLOS ONE

Additional Editor Comments (optional):

Reviewers' comments:

Reviewer's Responses to Questions

**Comments to the Author**

1. If the authors have adequately addressed your comments raised in a previous round of review and you feel that this manuscript is now acceptable for publication, you may indicate that here to bypass the “Comments to the Author” section, enter your conflict of interest statement in the “Confidential to Editor” section, and submit your "Accept" recommendation.

Reviewer #3: All comments have been addressed

2. Is the manuscript technically sound, and do the data support the conclusions?

Reviewer #3: Yes

3. Has the statistical analysis been performed appropriately and rigorously? 

Reviewer #3: Yes

4. Have the authors made all data underlying the findings in their manuscript fully available?

Reviewer #3: Yes

5. Is the manuscript presented in an intelligible fashion and written in standard English?

Reviewer #3: Yes

6. Review Comments to the Author

Reviewer #3: All my queries have been addressed by the authors. The paper can now be accepted for publication in the journal.

7. PLOS authors have the option to publish the peer review history of their article (what does this mean?). If published, this will include your full peer review and any attached files.

Reviewer #3: No

---

## [Editor Report · Acceptance letter]

15 May 2024

PONE-D-23-27070R2 

PLOS ONE

Dear Dr. Min, 

I'm pleased to inform you that your manuscript has been deemed suitable for publication in PLOS ONE. Congratulations! Your manuscript is now being handed over to our production team.

Kind regards, 

on behalf of

Prof Rehana Naz 

Academic Editor

PLOS ONE